# Learning to See Before Seeing: Demystifying LLM Visual Priors from Language Pre-training

**Junlin Han**[1,2]*, **Shengbang Tong**[1], **David Fan**[1], **Yufan Ren**[1], **Koustuv Sinha**[1],
**Philip Torr**[2] , **Filippos Kokkinos**[1]
[1]Meta Superintelligence Labs     [2]University of Oxford
`junlinhan@meta.com`

## Abstract

Large Language Models (LLMs), despite being trained on text alone, surprisingly develop rich visual priors. These priors allow latent visual capabilities to be unlocked for vision tasks with a relatively small amount of multimodal data, andto perform symbolic visual generation tasks without ever having seen an image. Through systematic analysis, we reveal that visual priors—the implicit, emergent knowledge about the visual world acquired during language pre-training—are composed of separable perception and reasoning priors with unique scaling trends and origins. We show that an LLM's latent visual reasoning ability is predominantly developed by pre-training on reasoning-centric data (*e.g.*, code, math, academia) and scales progressively. This reasoning prior acquired from language pre-training is transferable and universally applicable to visual reasoning. In contrast, the perception prior emerges more diffusely from broad corpora, and perception ability is more sensitive to the vision encoder and visual instruction tuning data. In parallel, text describing the visual world proves crucial, though its performance impact saturates rapidly. Leveraging these insights, we propose a data-centric recipe for pre-training vision-aware LLMs and verify it in 1T token scale pre-training. Our findings are grounded in over 100 controlled experiments consuming 500,000 GPU-hours, spanning the full MLLM construction pipeline—from LLM pre-training to visual alignment and supervised multimodal fine-tuning—across five model scales, a wide range of data categories and mixtures, and multiple adaptation setups. Along with our main findings, we also propose and investigate several hypotheses, and introduce a Multi-Level Existence Bench (MLE-Bench) to facilitate future research. Together, this work provides a new way of deliberately cultivating visual priors from language pre-training, paving the way for the next generation of multimodal LLMs.

We recommend a visit to our `project page` for an interactive reading.

## 1 Introduction

A compelling phenomenon has emerged at the forefront of AI research: Large Language Models (LLMs), despite being trained exclusively on vast corpora of text, appear to develop profound priors about the visual world. This latent capability is paradoxical, suggesting that the statistical patterns within language might be rich enough to encode fundamental principles of vision, from object properties to spatial relationships, without ever observing a single image. This emergent visual prior presents in several surprising and powerful ways:

**(1) Programmatic visual knowledge:** LLMs possess a rich visual knowledge, enabling them to generate executable code that renders complex 2D and 3D scenes, from objects to spatial layouts (Sharma et al., 2024; Ge et al., 2025; Ashutosh et al., 2025). This demonstrates a grasp of visual concepts without ever seeing a single image. The resulting synthetic data is of sufficient quality to pre-train standard vision models for successful generalization to real-world images (Sharma et al., 2024).

---

*Project lead

**(2) Data-efficient visual adaptation:** LLMs are highly efficient for visual adaptation. With a vision encoder, high-level reasoning emerges from instruction tuning on a small scale of image-text pairs, bypassing the need for massive multimodal pretraining (Alayrac et al., 2022; Liu et al., 2023a; Li et al., 2023; Grattafiori et al., 2024; Tong et al., 2024a; Bai et al., 2025b). This data-efficient instruction tuning extends to unified models, where visual generation is unlocked with minimal data (Tong et al., 2024b). Furthermore, this efficiency enables adaptation to low-level visual tasks using vision-only data (Zheng et al., 2024; Du et al., 2025), proving that an LLM's reasoning framework can function independently of cross-modal alignment.

**(3) LLMs as strong vision encoders:** The learned representations of LLMs can directly benefit pure vision tasks without language (Kumar et al., 2024; Pang et al., 2024; Lai et al., 2024; Bai et al., 2025a). When repurposed as visual encoders, the transformer layers of LLMs offer competitive performance on image classification, segmentation, and video understanding, even surpassing vision-specific backbones (Pang et al., 2024). These findings suggest that the hierarchical abstraction and long-range dependency modeling intrinsic to LLMs are not modality-specific, but rather capture general-purpose computational motifs that are well-suited to processing visual signals. This is also shown in neuron-level studies, which have identified multimodal neurons within LLMs that respond to the same abstract concept regardless of whether it is presented through text or vision (Schwettmann et al., 2023; Pan et al., 2023; Verma et al., 2024).

Collectively, these phenomena are not isolated curiosities; they point toward a deeper principle of representation learning. They lend strong empirical support to the Platonic Representation Hypothesis (Huh et al., 2024b; Jha et al., 2025), which posits that as models scale across diverse data and tasks, their latent representations—whether trained on text or images—converge toward a shared, underlying statistical model of reality. In this view, text and images are different "projections" or "shadows" of the world, and a powerful enough model can learn the structure of the world itself from any single projection. The visual priors in LLMs, therefore, may be a direct consequence of them recovering this unified internal world model from text alone.

These observations motivate a systematic investigation into the visual priors that LLMs acquire from language pre-training. Rather than a hand-crafted bias or a Bayesian prior distribution, we frame visual priors as implicit knowledge or prior vision capabilities encoded in LLMs, whose primary effect is to grant both enhanced capability for vision tasks and greater ease of transfer to vision. We seek to determine their origins, dissect whether they form a uniform block of knowledge or are composed of distinct, separable abilities, and explore how they can be leveraged to build more capable MLLMs.

Our work presents the first systematic investigation into the nature and origins of visual priors in the pre-training of LLMs and shows three key contributions: **(1): Structure of visual priors.** We establish that visual priors can be decomposed into perceptual and reasoning components. **(2): Source of visual priors.** We identify that the model's latent visual reasoning is predominantly cultivated by and scales progressively with reasoning-centric data, whereas its perception ability emerges more diffusely from broad, diverse data. **(3): Vision-aware language pre-training.** We propose a pre-training data-mixing strategy that strategically balances reasoning-centric and visually descriptive text to deliberately cultivate powerful visual priors for training LLMs that can result in stronger multimodal performance.

## 2 PROBLEM FORMULATION

In this section, we very briefly introduce our default training and evaluation settings. More details are available in Appendix B.

**Training protocol.** We pre-train Llama-3-style language models (340M to 13B parameters) with a default 3B parameter model trained on 30B tokens. Training data is composed of 16 sources. We then adapt these LLMs into MLLMs using a two-stage process: 1) visual alignment with an MLP projector, and 2) supervised fine-tuning on a curated mix of vision-language and language-only instruction data.

**Evaluation protocol.** Our evaluation is twofold. For LLMs, we measure perplexity for language modeling and average accuracy across eight benchmarks. For MLLMs, we follow Cambrian-1 and establish a diverse evaluation suite comprising 16 public benchmarks, we assess performance

General

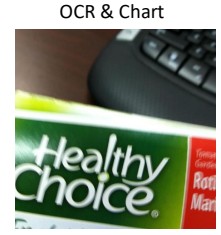

Is the dirty ground made of plastic?

Knowledge

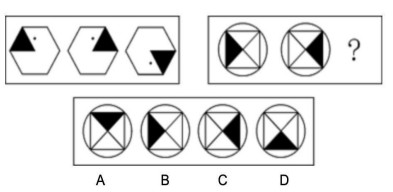

When a lightning flash of mean current $I$ and time $t$ occurs, the electric field strength between the cloud and Earth is reduced by how much?

OCR & Chart

What brand name is visible in the white text inside the green square?

Vision-Centric

Which one picture follows the same pattern established by the previous pictures?

Figure 1: **VQA evaluation question examples for each category.**

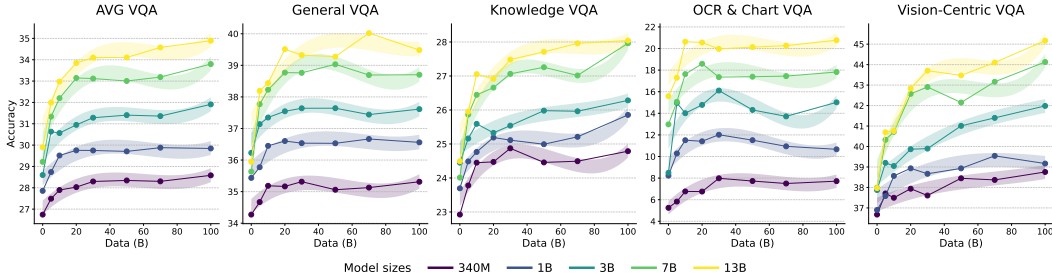

Figure 2: **Impact of model and data sizes.** The plots illustrate the performance of MLLMs, built upon LLMs of five different sizes (340M to 13B parameters), as a function of the amount of `web-crawl` pre-training data (0B to 100B tokens). The general trend shows that performance improves with both increasing model size and data volume, but the scaling behavior differs across task categories.

on 16 benchmarks grouped into four categories: General, Knowledge, OCR & Chart VQA, and Vision-Centric. Figure 1 presents example questions representing each category.

Additionally, we measure the representational alignment between language and vision modalities using a kernel-based similarity metric (Huh et al., 2024a).

## 3 DEMYSTIFYING LLM VISUAL PRIORS: STUDIES AND FINDINGS

This section presents our main results and findings. We first conduct a series of controlled experiments to systematically deconstruct the origins of LLM visual priors. Each subsection details its specific experimental setup or analytical approach, followed by the results and key findings.

### 3.1 IMPACT OF MODEL AND DATA SIZES.

> ***Finding 1:*** VQA performance scales positively with model and data size. However, this scaling is not uniform across all visual abilities.

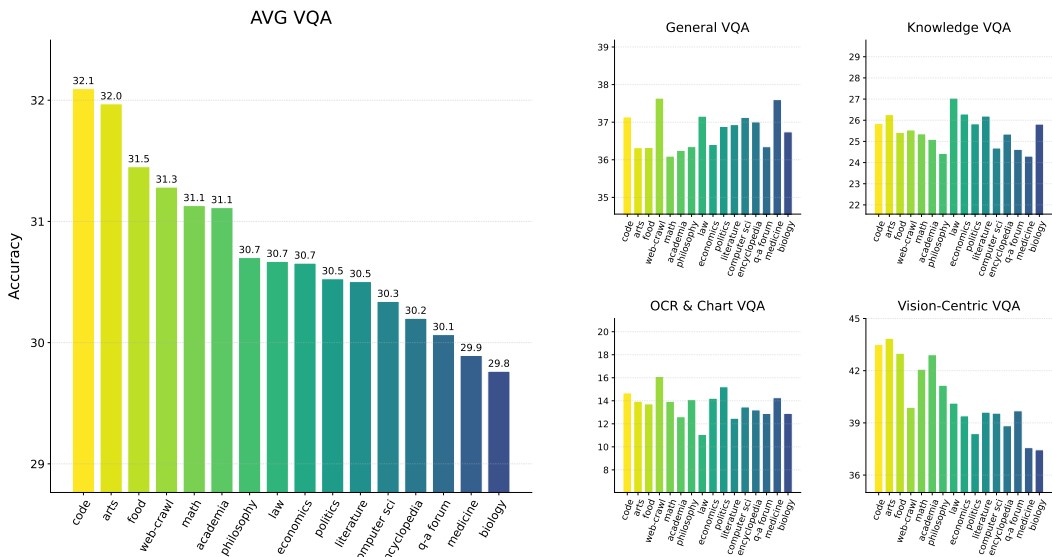

Figure 3: **Impact of pre-training data sources.** The bar charts illustrate the downstream VQA performance of MLLMs built upon a 3B parameter LLM, where each LLM was pre-trained on 30B tokens from a single, specific data source. The plots show that performance varies significantly depending on the pre-training sources.

We begin our analysis by investigating the fundamental impact of scale. To study how model size and pre-training data volume influence downstream multimodal capabilities, we perform a set of experiments to pre-train five LLMs of varying sizes (340M, 1B, 3B, 7B, and 13B parameters). Each model size was trained on eight different scales of data, ranging from 0B to 100B tokens. The training dataset is `web-crawl` for all experiments.

As illustrated in Figure 2, both model sizes and pre-training data sizes generally lead to stronger downstream multimodal performance. This holds true for the overall average VQA. However, a closer look at the different VQA categories reveals significant nuances. Performance on General VQA and Knowledge VQA demonstrates a similar scaling trend, consistently improving with both model and data size. In sharp contrast, OCR & Chart VQA is far more sensitive to model size than data volume; the performance gap between models is significantly wider. Meanwhile, Vision-Centric VQA also presents a unique pattern where the largest models benefit disproportionately from more data, while smaller models plateau much earlier. These divergent scaling patterns across different abilities demonstrate different visual abilities do not scale uniformly, but instead possess different properties that govern how they benefit from increased model and data size.

## 3.2 IMPACT OF PRE-TRAINING DATA SOURCES.

> ***Finding 2:*** Specific categories of language pre-training data can enhance certain visual capabilities in the resulting MLLM.

Having characterized the effects of data and model scales, we now transition our analysis to the composition of the data itself. To investigate the role of different pre-training sources, we fix the model size to 3B parameters and the total training data volume to 30B tokens. We then pre-train 16 distinct models, each trained exclusively on data from one of the 16 sources outlined in our pre-training sources (*e.g.*, `academia`, `biology`, `code`, etc.). This setup allows us to attribute performance variations directly to the specific data source used for pre-training.

As illustrated in Figure 3, the results reveal a significant variance in downstream multimodal performance depending on the pre-training data source. This divergence suggests that different categories of text data contribute to distinct and non-uniform visual priors. Notably, strong performance on Vision-Centric VQA tasks is highly correlated with two types of data: reasoning-centric (e.g., `code`,

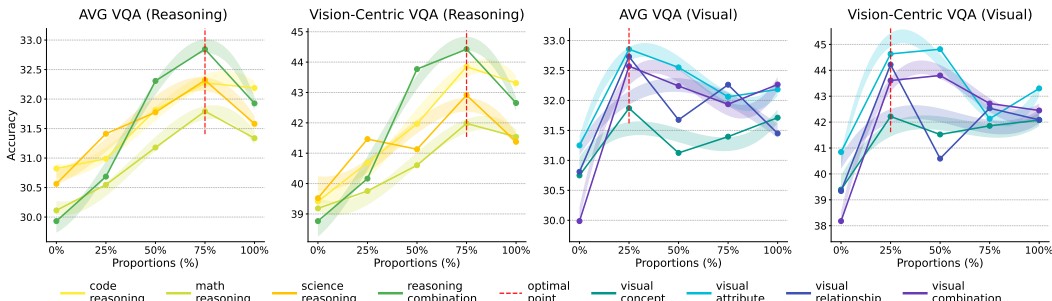

Figure 4: **Impact of reasoning-centric and visual data categories and proportions.** The plots illustrate how varying the proportion of specific data categories in the pre-training mix affects downstream VQA performance.

`mathematics`, `academia`) and corpora rich in visual world descriptions (e.g., `arts`, `food`). The top-performing models in Vision-Centric VQA, all scoring above 42%, are trained on these specific sources. This finding motivates a more granular analysis in the next section.

### 3.3 IMPACT OF REASONING AND VISUAL DATA CATEGORIES AND PROPORTIONS.

> ***Finding 3:*** A small amount of data about the visual world is crucial, but its contribution saturates quickly; in contrast, increasing the proportion of reasoning data in the pre-training mix progressively enhances visual abilities, with gains observed up to a 75% ratio.

Findings in the previous section show that reasoning-centric categories and categories related to the visual world were the most potent drivers of downstream visual capabilities. To dissect this phenomenon further, we focus our next set of experiments specifically on these two domains.

The reasoning-centric data was partitioned into `code reasoning`, `math reasoning`, `science reasoning`, and a `reasoning combination` category, which aggregates the three aforementioned categories. Concurrently, we define four categories for data related to the visual world: `visual concept`: Text naming visual entities like objects, people, places, and scenes. `visual attribute`: Descriptions of visual properties such as color, shape, texture, and style. `visual relationship`: Language detailing spatial arrangements or part-whole connections. `visual combination`: A combination of all three visual categories.

We begin by creating a data pool of approximately 300B tokens, comprising all sources used in Section 3.2. To partition this corpus into these categories, we employ a 32-B LLM (Yang et al., 2025a) to classify the text into finer-grained visual world and reasoning categories. Detailed classification settings and results are presented in the Appendix F.

With this fine-level categorization, we conduct a series of controlled mixing experiments to study how varying the proportion (mixing ratio) of these data types affects the final MLLM's performance. For each category, we train five separate models, systematically varying its proportion in the data mixture to 0%, 25%, 50%, 75%, and 100%. The remainder of the data for each run is drawn from a proportional mix of all other data within the 300B token pool, ensuring the total training volume is held constant at 30B tokens.

As shown in Figure 4, the results reveal a critical divergence in how visual world and reasoning data categories contribute to visual priors. The impact of reasoning-centric data is profound and progressive, with performance scaling steadily up to a 75% proportion. The contribution from data explicitly describing the visual world saturates quickly; a small initial amount seems to be crucial, but further increases yield diminishing returns.

## 3.4 DERIVING A DATA MIXTURE FOR MORE VISION-AWARE LLMS.

> **Finding 4:** Maximizing MLLM VQA performance is best achieved by pre-training on a data mixture heavily skewed towards reasoning-centric content but with necessary vision world knowledge. The balance point between language and vision proficiency is reached via a calibrated data mixture between language-favorable and vision-favorable.

| Data Ratio | | Avg VQA | Data Ratio | | Avg VQA |
|---|---|---|---|---|---|
| rea | vis | | rea | vis | |
| 50 | 5 | 30.7 | 55 | 5 | 30.9 |
| | 10 | 31.3 | | 10 | 31.7 |
| | 15 | 31.8 | | 15 | 32.2 |
| 60 | 5 | 31.9 | 65 | 5 | 32.0 |
| | 10 | 32.4 | | 10 | 32.2 |
| | 15 | **32.7** | | 15 | 32.5 |
| | 20 | 32.5 | | 20 | 32.1 |
| | 25 | 32.4 | | 25 | 31.9 |
| | 30 | 31.6 | | 30 | 31.4 |
| 70 | 5 | 31.9 | 75 | 5 | 31.6 |
| | 10 | 32.3 | | 10 | 31.5 |
| | 15 | 32.6 | | 15 | 32.4 |
| 80 | 5 | 31.5 | 85 | 5 | 31.2 |
| | 10 | 32.4 | | 10 | 31.6 |
| | 15 | 32.2 | | 15 | 31.8 |

Table 1: **Grid Search for a vision-favorable data mixture.** Results from pre-training a 3B parameter LLM on 30 distinct data blends, each totaling 30B tokens. The table explores how varying the proportions of reasoning-centric (`rea`) and visual-world (`vis`) data affects various capabilities.

Building on these findings, our objective is to derive a single, practical data mixture that not only excels on language tasks but also serves as a powerful foundation for MLLMs. Our approach proceeds in three stages. First, we determine a vision-favorable blend using our 300B token pool to establish a target. For the subsequent, more practical stages of our analysis, we then narrow our focus to six primary data categories: `web-crawl`, `encyclopedia`, `academia`, `literature`, `math`, and `code`. Within this practical set of sources, we then identify a language-favorable mixture (second), and finally, derive a balanced mixture by interpolating between these two optima (third). Since our goal also involves downstream VQA performance, we adapt the most direct approach of using a grid search over mixture ratios.

**Vision-favorable mixture.** We first aim to identify a data mixture that excels at visual tasks. To do so, we conduct a grid search over the proportions of reasoning-centric and visual-world data drawn from our 300B token pool. Specifically, we perform a grid search across 24 data blends constructed by sampling from a space where the `reasoning combination` ranges from 50% to 85% and the `visual combination` ranges from 5% to 30%, following the conclusions drawn from Section 3.3. The comprehensive results of this search are presented in Table 1.

From this search, we find that the best-performing models for downstream MLLM tasks emerge from a mixture containing approximately 60% reasoning and 15% visual content. Results show a powerful visual foundation is not built by simply maximizing exposure to visual descriptions, but by establishing a strong reasoning faculty, which is then grounded by a smaller amount of visual world knowledge. This experiment is performed on our 300B token pool, provides a target ratio for maximizing VQA performance. In the next part, we proceed to a more practical experiment at the data source level, guided by this result.

**Language-favorable mixture.** We begin by establishing a language-favorable mixture that achieves the best performance on our language task suite. Guided by recent literature (Shukor et al., 2025a; Ge et al., 2024; Ye et al., 2024) and empirical testing over 10 experiments, we identify this as a mix of 50% `web-crawl`, 2.5% `encyclopedia`, 2.5% `academia`, 20% `literature`, 5% `math`, and 20% `code`. This blend, designated as mix0 in Table 2, serves as our baseline for strong language proficiency, achieving the highest text accuracy (53.0%) and the best perplexity (13.46).

**Balanced mixture.** To reconcile these two objectives, we seek a single, balanced mixture that offers strong performance across both modalities. We achieve this by performing a series of interpolation experiments, detailed as mix0 through mix10. We shift the data composition from our language-favorable baseline (mix0) towards an endpoint representing the vision-favorable blend (approximated by mix9 and mix10). To ensure stabilized results, each model in this series is trained for 50B tokens.

| Recipe | | Data Source Mixture (%) | | | | | | | | Performance Metrics | | | Overall Rank |
|---|---|---|---|---|---|---|---|---|---|---|---|---|---|
| | | web-crawl | encyclopedia | academic | literature | math | code | reasoning | visual | t-acc (%) | ppl (↓) | v-acc (%) | |
| mix0 | | 50.0 | 2.5 | 2.5 | 20.0 | 5.0 | 20.0 | 33.1 | 21.7 | **53.0** | **13.46** | 32.4 | 5 |
| mix1 | language | 48.3 | 3.4 | 2.9 | 17.0 | 5.8 | 22.5 | 36.2 | 20.6 | 52.8 | 13.48 | 32.4 | 4 |
| mix2 | | 46.7 | 4.3 | 3.3 | 14.0 | 6.7 | 25.0 | 39.4 | 19.4 | 52.6 | 13.51 | 32.6 | 8 |
| mix3 | | 45.0 | 5.2 | 3.8 | 11.0 | 7.5 | 27.5 | 42.6 | 18.2 | 52.5 | 13.56 | 32.9 | 9 |
| mix4 | | 43.3 | 6.1 | 4.2 | 8.0 | 8.3 | 30.0 | 45.7 | 17.1 | 52.4 | 13.62 | 32.7 | 10 |
| mix5 | | 41.7 | 7.1 | 4.6 | 5.0 | 9.2 | 32.5 | 48.9 | 16.0 | 52.6 | 13.57 | 33.0 | 6 |
| mix6 | | 40.0 | 8.0 | 5.0 | 2.0 | 10.0 | 35.0 | 52.0 | 14.8 | 52.7 | 13.52 | 33.3 | **1** |
| mix7 | | 36.5 | 7.0 | 7.5 | 2.0 | 11.5 | 35.5 | 55.5 | 14.4 | 52.5 | 13.56 | 33.1 | 3 |
| mix8 | vision | 33.0 | 6.5 | 9.5 | 2.0 | 12.0 | 37.0 | 57.2 | 14.0 | 52.7 | 13.52 | 33.2 | 2 |
| mix9 | | 29.5 | 6.0 | 11.5 | 2.0 | 12.5 | 38.5 | 59.0 | 13.6 | 52.3 | 13.71 | 33.2 | 7 |
| mix10 | | 26.0 | 5.5 | 12.5 | 2.0 | 13.0 | 41.0 | 61.3 | 13.3 | 52.1 | 13.88 | **33.4** | 11 |

Table 2: **Deriving a data mixture for more vision-aware LLMs.** This table details a series of 11 data mixtures, from mix0 (language-favorable blend) to mix10 (approximating the vision-favorable blend), all trained on a 3B-parameter LLM with 50B tokens. The results highlight a trade-off, with mix6 emerging as the most balanced mixture, achieving top-ranked overall performance by improving visual capabilities without a significant drop in language proficiency.

The performance metrics in Table 2 reveal the expected trade-off: as the mixture becomes more reasoning-centric, vision accuracy (v-acc) generally improve, while language proficiency (t-acc and ppl) shows a slight decline. Our analysis identifies mix6 as the balanced mixture, achieving the highest overall rank. Mixtures in its vicinity (e.g., mix5, mix7, mix8) also achieve high rankings. This demonstrates that a carefully calibrated data mixture can cultivate powerful visual priors without substantially compromising core language abilities.

### 3.5 THE STRUCTURE AND ORIGIN OF LEARNED VISUAL PRIORS.

> ***Finding 5:*** The learned visual prior is not a single entity but decomposes into at least a perception prior and a reasoning prior with different origins.

We now synthesize our previous results to investigate the internal structure of the visual prior. We conceptualize the visual prior as a collection of distinct abilities, each measured by one of our four VQA categories.

**Internal structure of visual priors.** We aggregate the performance data across all 105 3B models from our previous experiments, encompassing variations in data sources, mixing ratios, and training scales. We then compute the Spearman correlation matrix across the four VQA performance categories to identify which abilities scale together and which diverge.

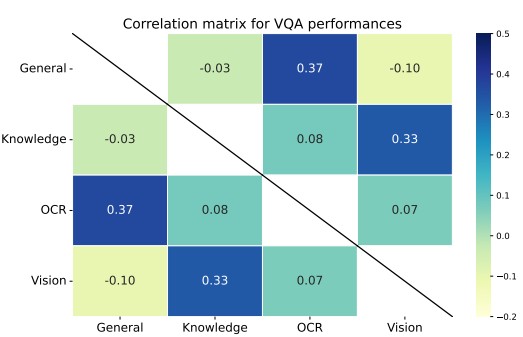

Figure 5: **Correlation matrix for VQA performances.** The matrix reveals two loosely-coupled skill clusters: one axis for perception (General/OCR) and another for reasoning (Knowledge/Vision-Centric).

The results in Figure 5 suggest a potential internal structure within the visual prior, hinting at a separation into at least two distinct types of abilities. We observe a moderate correlation (0.37) between General and OCR performance. This connection seems to point towards a perception prior, as success in both categories relies heavily on the model's perceptual acuity—the ability to accurately process raw visual input—rather than complex, multi-step reasoning.

In contrast, we find another moderate correlation (0.33) between the Knowledge and Vision-Centric tasks. This link appears to emerge because both categories often require abstract inference that goes beyond simple perception. For instance, the Knowledge category demands multi-step reasoning to solve complex scientific or mathematical problems, while Vision-Centric tasks include challenges like visual IQ puzzles, object counting, and correspondence matching, necessitate a blend of perception and reasoning, often with a heavier reliance on the latter. The correlation matrix also reveals very weak, or even slightly negative, correlations between these two

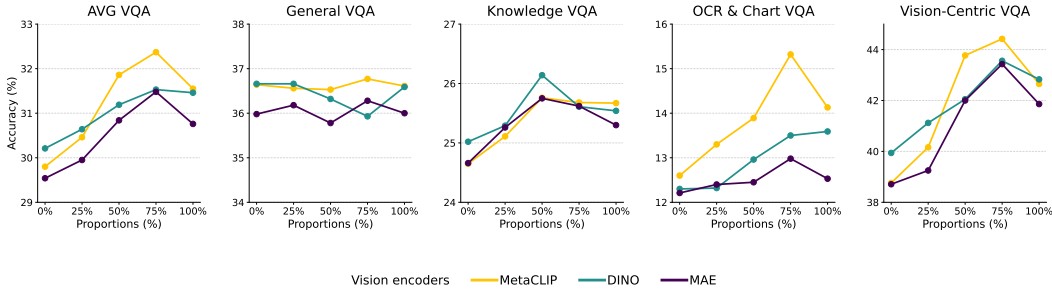

Figure 6: **Universality of the learned visual priors.** The plots show the VQA performance of MLLMs built using three distinct vision encoders based on the proportion of reasoning-centric data used in the LLM's pre-training mix. Despite differences in their absolute performance, all three configurations show a consistent improvement on reasoning-heavy tasks as the LLM's reasoning pre-training proportion increases, similar to trends observed before.

groups (perception-heavy vs. reasoning-heavy)[1]. This lack of a strong positive correlation raises the possibility that these are largely independent abilities, potentially stemming from loosely-coupled priors within the LLM's representation. Our observations on the separability of these visual priors in MLLMs align with and extend the findings of recent research Chen et al. (2025), which identified a similar dissociation through parameter merging.

**Different origins of priors.** The statistical independence of these two priors implies they are cultivated through different mechanisms. As our analysis in Section 3.2 and Section 3.3 demonstrated, the reasoning prior is from reasoning-centric data and can be predictably enhanced by increasing the proportion of reasoning-centric data.

In contrast, the origins of the perception prior appear more diffuse. A signal comes from our single-source experiments (Section 3.2), where `web-crawl` data yields the best performance on General and OCR tasks. However, `web-crawl` is an extremely general category, and no other, more specific data category consistently boosts perceptual abilities. This mixed effect suggests the perception prior may be a general byproduct of large-scale language modeling, emerging from the sheer diversity of language rather than a specific category.

To further investigate this perception prior and characterize its properties more directly, we introduce a multi-level existence benchmark (MLE-Bench) designed to assess pure perception abilities across multiple levels. In Section A.1, we use this benchmark to test our hypothesis that the perception prior is scale-dependent and is most pronounced for small and medium-sized objects. Further details on the benchmark's construction are presented in Appendix I. We also present an evaluation of some MLLMs on MLE-Bench, revealing that they exhibit varying levels of ability in perceiving objects of different sizes. Detailed results and analysis are available in Appendix J.

### 3.6 DECONSTRUCTING MULTIMODAL ABILITIES: VISION OR LANGUAGE.

> ***Finding 6:*** Visual reasoning is primarily shaped by reasoning prior acquired from language pre-training; perception is more dependent on post-training (visual instruction tuning).

Here, we conduct further analysis to first verify the universality of learned visual priors and then deconstruct the source of different multimodal abilities, distinguishing between those inherited more from the LLM and those acquired more from the visual instruction tuning.
**Universality of the learned visual priors.** To test the general influence of the visual prior, we apply two more vision encoders (DINOv2-G (Oquab et al., 2023) and MAE-H (He et al., 2022)) other than our default MetaCLIP-B/16. We pair these with LLMs pre-trained on varying proportions of our `reasoning combination` data category, from 0% to 100%.

---

[1]This categorization into perception-heavy and reasoning-heavy tasks is a conceptual simplification intended to facilitate our analysis. The boundary between perception and reasoning is not always clear.

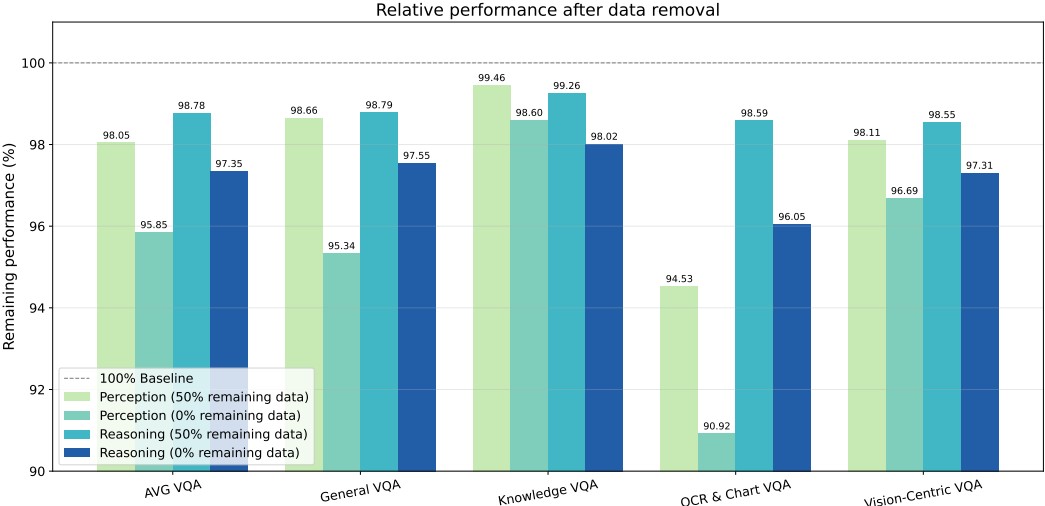

Figure 7: **Step-wise effects of perception and reasoning visual-instruction tuning data removal.** The chart shows the remaining performance (%) on Avg VQA and per-category VQA (x-axis) relative to a 100% baseline. The bars show performance after ablating perception or reasoning instruction data in stages (removing 50% and then 100% of the data).

As illustrated in Figure 6, the results reveal a dual-faceted pattern. Firstly, they confirm the universality of the reasoning prior. For reasoning-heavy tasks, all three vision encoder configurations exhibit a nearly identical, strong upward trend in performance as the proportion of reasoning data in the LLM's pre-training increases. This demonstrates that the visual reasoning prior cultivated in the LLM is a foundational, modality-agnostic prior that benefits the multimodal system regardless of the specific vision encoder used. In contrast, the perception prior lacks this universality. The performance trends for perception-oriented tasks are more inconsistent across the different vision encoders.

**Source of abilities, from visual priors or visual instruction tuning.** Second, we conduct targeted studies to determine whether perception and reasoning skills originate primarily from the LLM's visual priors or the subsequent visual instruction tuning stage. We use an MLLM to classify our Cambrian-7M dataset that contains 5M text-image pairs into these two categories, resulting in 1.8M perception and 0.6M reasoning data, and the remaining 2.6M data as others. Further classification details are provided in the Appendix G.

We partition our instruction-tuning data into perception, reasoning, and other categories and trained five tuning configurations that ablate perception and reasoning data in stages (100% → 50% → 0%) while leaving other data unchanged. The results, presented in Figure 7, show two observations: (1) reducing perception-targeted tuning produces the largest performance drops on perception-heavy benchmarks and modest drops on reasoning tasks; (2) removing reasoning-targeted tuning causes only small incremental drops on perception tasks and modest drops on reasoning tasks.

Together, results in this section show two mechanisms. First, the LLM encodes a robust, transferable visual reasoning prior primarily via language pre-training; this prior benefits reasoning-centered VQA across different vision encoders. Second, perception performance depends more on vision-encoder characteristics and on subsequent supervised visual instruction tuning.

In addition to our main findings, we investigate the universality of reasoning priors acquired during pre-training. Our experiments show that this reasoning ability is highly transferable and modality-agnostic, as detailed in Appendix A.2. Furthermore, in Appendix A.3, we study whether language data structures drive representational alignment with vision, finding that they can indeed partially drive this alignment.

## 4 SCALING UP AND TRAINING A VISION-AWARE LLM

### 4.1 SETTINGS AND MODELS

Building upon our findings, we scale up our approach to validate our findings and develop a vision-aware LLM on a larger-scale. The goal is to test whether the principles identified in our controlled, smaller-scale studies hold true when applied to larger training runs. To this end, we pre-train two 7B parameter LLMs, each on 1T tokens, based on the two data mixtures identified previously: **Language-favorable model**: Following the mix0 mixture, which is the best-performing blend for pure language tasks. **Balanced model**: Based on the mix6 recipe, our proposed balanced mixture is designed to deliberately cultivate strong visual priors without compromising language proficiency.

| Model | Language | | Vision | | | | |
|---|---|---|---|---|---|---|---|
| | ppl | avg acc | General | Knowledge | OCR&Chart QA | Vision-Centric | Overall |
| Language-Favorable | 8.72 | 0.647 | 46.92 | 28.35 | 21.49 | 46.31 | 37.32 |
| Balanced | 7.49 | 0.655 | 49.59 | 29.02 | 23.63 | 46.59 | 38.64 |

Table 3: **Performance comparisons of the Language-Favorable and Balanced models across both language and vision-language benchmarks.** The table summarizes key language metrics (perplexity and accuracy) and provides average scores for a suite of vision tasks.

### 4.2 RESULTS

As shown in Table 3, the Balanced model, pre-trained with balanced recipe, exhibits competitive language proficiency. Notably, it achieves a lower (better) average perplexity of 7.49 compared to the Language-favorable model's 8.72, while also maintaining a slightly higher average accuracy (0.655 vs. 0.647). An interesting dynamic observed during pre-training was that the Balanced model's language performance initially lagged behind the Language-favorable model, beginning to surpass it after approximately 600B tokens. This may suggest that when the pre-training token volume is sufficiently large, the benefits from reasoning-related tokens can be more effectively unleashed when grounded in a substantial amount of world knowledge.

On VQA benchmarks detailed, the Balanced model consistently outperforms the Language-Favorable model in most of the benchmarks, achieving a higher overall VQA average (38.64 vs. 37.32). This confirms that the deliberate pre-training on a data mixture rich in reasoning and visual world text successfully imbues the LLM with stronger visual priors in a larger scale.

We also introduce Blind Visual Instruction Tuning, a trick that serves the MLLM community as both a practical tool for visual adaptation and a probe for revealing how models "hack" visual tasks with language (see Appendix K). Together with this trick, we find that many MLLMs, even state-of-the-art ones, fail to recognize the absence of an image input. This leads to hallucinations in visual question answering scenarios without a visual context. A study of this behavior is presented in Appendix L.

## 5 CONCLUSION

This work has undertaken a systematic deconstruction of the visual priors that LLMs acquire from text-only pre-training. Through a series of controlled experiments manipulating data composition, we moved beyond observing the phenomenon of vision priors to interrogating its fundamental drivers. Our investigation provides a data-centric roadmap for developing multimodal systems, shifting the paradigm from serendipitous emergence to the deliberate cultivation of visual capabilities.

Looking forward, we hope this research encourages a paradigm where LLM development is more considerate of vision and multimodality, prompting the cultivation of visual priors from the earliest stages of pre-training. We also hope it inspires a deeper investigation into the fundamental correlations between cross-modal representations, contributing to a more unified understanding of how knowledge is structured across modalities.

## THE USE OF LARGE LANGUAGE MODELS (LLMS)

LLMs were used to aid and polish paper writing.

## ETHICS STATEMENT

This research was conducted with a commitment to responsible AI practices. The ethical implications of this research are primarily related to the societal biases present in large-scale text corpora. A more detailed discussion of these considerations and the broader impact of our work is provided in the Broader Impact section in Appendix E.

## REPRODUCIBILITY STATEMENT

To support the reproducibility of our findings, we will publicly release our newly proposed Multi-Level Existence Bench (MLE-Bench). The detailed methodology for classifying our pre-training data is described in Appendix G and Appendix F. Furthermore, details regarding training protocols and evaluation procedures are provided in Appendix B.

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

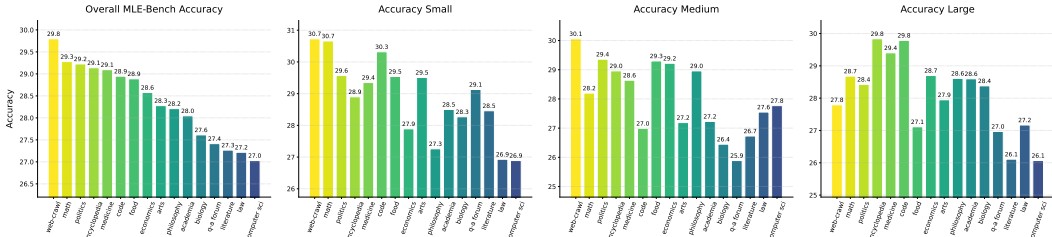

Figure 8: **Performance of MLLMs on the Multi-Level Existence Bench (MLE-Bench).** The left plot shows the overall accuracy for models pre-trained on 16 different single-source data types. Other plots detail performance on objects of varying relative sizes, from small (0-30% of image pixels) to medium (30-60%) to large (60-100%). The results demonstrate that pre-training on the broad and diverse `web-crawl` corpus is most effective in gaining perception prior, with its advantage being particularly pronounced for perceiving smaller objects.

## A DISCUSSION AND HYPOTHESES

This section transitions from empirical findings to a more speculative exploration of the underlying mechanisms of visual priors. The following subsections present three key hypotheses about the structure of the perception prior, the universal nature of reasoning, and the role of data structure in cross-modal alignment. These hypotheses are not presented as definitive conclusions but as frameworks for interpreting the results and for future research.

### A.1 IS THE PERCEPTION PRIOR MULTI-LEVEL? AN EVALUATION USING THE MLE-BENCH

> ***Hypothesis 1:*** The perception prior derived from diverse data exhibits scale-dependency, with its benefits being most pronounced for the perception of small and medium-sized objects.

Our previous analyses show that the perception prior is diffuse in origin, emerging most strongly from diverse data. This leads to a question about its internal structure: is this prior a uniform ability, or does it possess finer-grained characteristics?

To study this question, we introduce the Multi-Level Existence Bench (MLE-Bench), a benchmark designed to probe perception with greater precision. MLE-Bench consists of 4-choice questions about the existence of objects or scenes within an image. We categorize questions based on the target object's relative size, measured by the percentage of pixels it occupies. In total, MLE-Bench comprises 1,861 images, with a distribution of 732 questions for small objects (0-30%), 698 for medium objects (30-60%), and 431 for large objects (60-100%). This structure allows us to deconstruct "perception" into distinct, scale-dependent components. Further details on the benchmark's construction are presented in Appendix I. We also present an evaluation of common MLLMs on MLE-Bench, with detailed results and analysis available in Appendix J.

We evaluate our 16 single-source pre-trained models (from Section 3.2) on MLE-Bench, with the results presented in Figure 8. The 3B LLM model trained on web-crawl remains the top performer overall, confirming that data diversity is key for perception prior. Its advantage is most pronounced for small-to-medium objects (0-60% pixel range), where it establishes a clear lead over models trained on other data sources. In contrast, for large objects that dominate the visual scene, this performance gap diminishes significantly.

These results indicate that the perception prior is indeed scale-dependent. A possible explanation is that diverse, unstructured text like web-crawl contains a vast vocabulary describing a wide array of entities, including smaller, often overlooked details within a larger scene. This textual richness forces the model to learn representations sensitive to fine-grained visual concepts, a capability less critical when identifying large, obvious objects. This finding refines our understanding of the perception prior, revealing that it is not a uniform faculty.

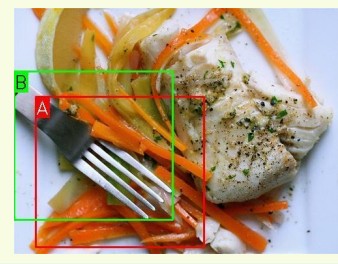

Figure 9: **Qualitative impact of reasoning-centric data on visual reasoning.** The figure displays the answers from five models—pre-trained with 0% to 100% `code reasoning` data—to a visual question requiring the application of a specific rule. Answers show a clear improvement in reasoning quality: the model with 0% `code reasoning` provides a simplistic justification, while the models with 75% and 100% `code reasoning` produce more detailed reasoning that correctly applies the definition from the prompt.

## A.2 IS REASONING A UNIVERSAL, CROSS-MODAL SKILL ALREADY ACQUIRED DURING PRE-TRAINING?

> **Hypothesis 2:** The reasoning capabilities an LLM acquires from text are fundamentally modality-agnostic. Language reasoning skills can be directly transferred to solve visual problems.

Our findings suggest a hypothesis: the reasoning capabilities an LLM acquires from text are not bound to the linguistic domain. We posit that by pre-training on reasoning-centric data, a model learns abstract, generalizable principles of logic, structure, and compositionality. This foundation is largely modality-agnostic, allowing the model to apply this faculty to other domains, including vision, since the reasoning process likely occurs within the language domain.

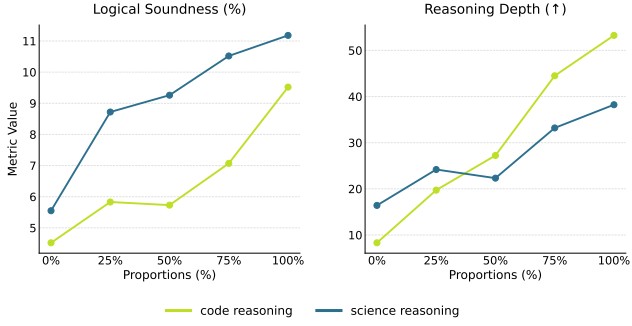

Figure 10: **Qualitative impact of reasoning-centric data on visual reasoning tasks.** The plot shows how varying the proportion of different reasoning-centric data categories in the pre-training mix impacts metrics of visual reasoning quality. Results indicate that more reasoning data leads to more coherent and detailed visual reasoning.

To verify this, we propose an experiment that directly probes the quality of the models' visual reasoning processes. For reasoning-focused VQA tasks (Knowledge and Vision-Centric), we switch the evaluation from prompting for a direct answer to answering with detailed explanations. This will require each model to produce a detailed explanation of its reasoning. We then use a separate LLM (Yang et al., 2025a) as a judge to evaluate the quality of these reasoning traces based on a clear rubric, assessing criteria such as: (1) Logical Soundness: The percentage of reasoning traces that are coherent and reasonable; and (2) Reasoning Depth: The average length of the reasoning trace measured by text count taken to reach the conclusion.

The results, presented in Figure 10, strongly support our hypothesis that the reasoning capabilities an LLM acquires from text are transferable to vision. We observe a clear trend: as the proportion of reasoning-centric data increases, the models generate visual reasoning that is both more logically sound and significantly longer. For instance, increasing the proportion of `code reasoning` data from 0% to 100% boosts Logical Soundness from 4.52% to 9.52% and more than sextuples the Reasoning Depth from 8.31 to 53.25. This demonstrates that the model is applying a general, abstract reasoning framework, learned from text, to solve visual problems. The particularly dramatic increase in Reasoning Depth for code-trained models may also reflect a stylistic transfer; pre-training on `code reasoning`, which is often structured in long, logically coherent sequences, likely predisposes the model to generate longer, more structured step-by-step explanations.

Figure 9 provides a qualitative example of this phenomenon. It showcases how models trained with more `code reasoning` data produce increasingly sophisticated and reasonable reasoning for a visual task. While the model with 0% `code reasoning` offers a simplistic justification, the model trained on 100% `code reasoning` provides a detailed, step-by-step explanation that correctly applies the abstract rule given in the prompt. This demonstrates that the model is applying a general, abstract reasoning framework, learned from text, to solve visual problems. Our conclusions here also reflect the results shown in prior work showing cross-lingual reasoning transfer (Yong et al., 2025), improvements in multimodal reasoning via post-training (Rastogi et al., 2025; Wei et al., 2025; Yang et al., 2025b; Liu et al., 2025; Chen et al., 2025), transfer of structured procedural knowledge from text to non-linguistic tasks (Ruis et al., 2024), and temporal reasoning transfer from text to video understanding (Li et al., 2024). We further demonstrate that reasoning abilities are highly modality-agnostic, to the extent that training solely on `code` can yield strong multimodal reasoning. Moreover, we show this transferability is not confined to post-training phases.

### A.3 DOES LANGUAGE DATA STRUCTURE DRIVE REPRESENTATIONAL ALIGNMENT WITH VISION?

> **Hypothesis 3:** The structural properties of language data can partially drive representational alignment with visual data.

An alternative, or perhaps complementary, hypothesis centers on the structural similarities between the data modalities themselves.

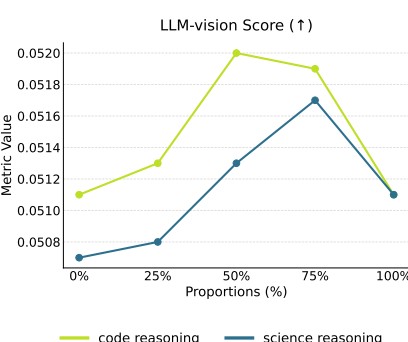

Figure 11: **Qualitative impact of reasoning-centric data on representation alignment.** The plot show how varying the proportion of different reasoning-centric data categories in the pre-training mix impacts metrics of cross-modal alignment. Results reveal that with the LLM-vision alignment score showing a generally positive but non-monotonic trend.

Data from domains like `code` and `mathematics` is inherently highly structured. It is governed by strict syntax, logical dependencies, and hierarchical compositions. Similarly, visual data is far from being a random collection of pixels. It is rich with its own structure: spatial relationships between objects, part-whole hierarchies, and the implicit rules of physics and geometry. We thus hypothesize that this shared structural foundation means that representations learned from structured text are intrinsically more similar to, and thus more readily transferable to, the visual domain.

To test this hypothesis, we analyze the LLM-vision alignment score across models trained with varying proportions of structured reasoning data. Our analysis, presented in Figure 11, reveals a clear but non-monotonic trend. As we increase the proportion of structured reasoning data, the alignment score generally improves, suggesting that learning from abstract structure fosters a more congruent latent space. However, this trend peaks at a 75% ratio before declining at 100%. This might be due to a model trained purely on reasoning data learning abstract structure but lacking the necessary vocabulary from other text types to effectively map it onto diverse visual concepts, thus hindering the final alignment. While the relationship is not linear, this analysis provides some signals for our

structural similarity hypothesis. It remains a compelling direction for future research to untangle the precise interplay between abstract structure and semantic grounding in forming cross-modal representations.

## B PROBLEM FORMULATION DETAILS

### B.1 TRAINING PROTOCOL.

**LLM pre-training setup.** We follow standard practices and pre-train a suite of decoder-only Transformer models that closely adhere to the Llama-3 architecture (Grattafiori et al., 2024), spanning five model scales: 340M, 1B, 3B, 7B, and 13B parameters. These models are trained for varying numbers of tokens at 0B, 5B, 10B, 20B, 30B, 50B, 70B, 100B, and up to 1T tokens. We use a tokenizer with a vocabulary size of approximately 32000. Training is performed using the AdamW optimizer (Loshchilov & Hutter, 2017) with a peak learning rate of $3 \times 10^{-4}$, following a cosine decay schedule and a warm-up over the first 1024 steps. All models are trained with a context length of 2048 tokens and an effective global batch size of 1024. We fix the model size to 3B parameters and the total training data volume to 30B tokens as our default setting.

**LLM pre-training data.** Our training data is composed of a diverse mixture of 16 sources, including `academic`, `arts`, `biology`, `code`, `computer science`, `economics`, `encyclopedia`, `food`, `law`, `literature`, `mathematics`, `medicine`, `philosophy`, `politics`, `q-a forum`, and `web-crawl`. Each source contains at least 50B tokens.

**MLLM adaptation setting.** We adopt a two-stage adaptation strategy following Cambrian-1 (Tong et al., 2024a) and Web-SSL (Fan et al., 2025), consisting of visual alignment and supervised fine-tuning. In the first stage, we train an MLP-based projector on top of a frozen vision encoder and language model to align visual features with the LLM. Unless otherwise specified, we use MetaCLIP-B/16 (Xu et al., 2023) as the default vision encoder. Extracted visual features are uniformly resized to a fixed length of 576 tokens. In the second stage, we perform supervised fine-tuning on a mixture of vision-language and language-only instruction data to enhance the model's multimodal instruction-following ability. Both the alignment and instruction tuning stages use the AdamW optimizer with a cosine learning rate schedule and linear warm-up, and models are trained for a single epoch. During alignment, we use a learning rate of $1 \times 10^{-3}$ with a warm-up ratio of 6%. For instruction tuning, the learning rate is set to $4 \times 10^{-5}$ with a 3% warm-up. Training is conducted with an effective global batch size of 512.

**MLLM adaptation data.** We adopt the Cambrian-1 and Web-SSL data pipeline, but with strategic data reductions to highlight the effect of vision priors. The initial alignment stage utilizes a 1M image-caption dataset, which is roughly 40% of the original dataset's size. This is followed by supervised fine-tuning on a curated 3.5M subset of the Cambrian-7M data. This subset is balanced with approximately 1.5M language-only and 2M vision-language paired instructions, resulting in a higher percentage of language-only instruction data than the original curation, as our models learn to follow language instructions during this phase.

For all experiments, we fix the random seed as 42. For MLLM adaptation, we use the same order for data loading in both alignment and instruction tuning stage to get stable results.

### B.2 EVALUATION PROTOCOL.

**LLM evaluation.** We conduct a comprehensive evaluation of our pre-trained models' language understanding and reasoning abilities. Following the benchmark suite used in Mamba (Gu & Dao, 2023) and GLA (Yang et al., 2023), we assess performance on two main fronts. For raw language modeling quality, we report the averaged perplexity (ppl) across Wikitext (Merity et al., 2016) and LAMBADA (Paperno et al., 2016). For reasoning, we evaluate zero-shot performance on a diverse suite of commonsense and question-answering tasks, including PIQA (Bisk et al., 2020), HellaSwag (Zellers et al., 2019), WinoGrande (Sakaguchi et al., 2021), ARC (Clark et al., 2018), Copa (Reddy et al., 2019), SciQA (Auer et al., 2023), OpenbookQA (Mihaylov et al., 2018), and

BoolQA (Clark et al., 2019). For a concise comparison, we report the unweighted averaged accuracy over all these benchmarks.

**MLLM evaluation.** To comprehensively assess the multimodal capabilities of our models, we follow Cambrian-1 and establish a diverse evaluation suite comprising 16 public benchmarks. We group these benchmarks into four key categories to isolate and probe the distinct components of the learned visual prior, ranging from fine-grained perception to abstract reasoning, and provide a holistic view of model performance:

- General: This category probes the model's ability to perform visual perception and connect it with commonsense knowledge, rather than complex, multi-step reasoning. It includes GQA (Hudson & Manning, 2019), MME (Fu et al., 2023), MMBench (Liu et al., 2024c), and SEED (Ge et al., 2023).

- Knowledge: This category evaluates the model's capacity to connect visual information with the world and perform multi-step reasoning to solve complex scientific or mathematical problems. It covers ScienceQA (Lu et al., 2022b), MMMU (Yue et al., 2024), AI2D (Hiippala et al., 2021), and MathVista (Lu et al., 2023).

- OCR & Chart VQA: This category focuses on fine-grained perception, specifically the ability to accurately read and interpret dense textual and structured data within images. It comprises TextVQA (Sidorov et al., 2020), ChartQA (Masry et al., 2022), and OCRBench (Liu et al., 2023b).

- Vision-Centric: This category mainly probes abstract visual reasoning and rough perception skills, requiring the model to perform tasks such as spatial and 3D understanding, object counting, and IQ tests. It uses benchmarks including RealWorldQA (xAI, 2024), Blink (Fu et al., 2024), COCO, ADE, and Omni3D. COCO (Lin et al., 2014), ADE (Zhou et al., 2019), and Omni3D (Brazil et al., 2023) are proposed as CV-Bench from Cambrian-1 (Tong et al., 2024a).

The overall average result is based on all benchmarks. We also report the averaged multimodal evaluation accuracy for each category. To fairly assess a model's core vision ability, independent of its instruction-following capabilities, we address a challenge: models, especially smaller ones, often embed correct answers within conversational text rather than providing a direct response. Our evaluation uses a robust parsing strategy to extract the intended answer from this free-form text. This approach ensures a reliable assessment of all models, including those without language pre-training but only visual instruction tuning, making our results resilient to variations in response formatting. More details about this parsing strategy are presented in Appendix H.

**LLM-vision alignment.** To quantify the representational convergence and similarity between language and vision modalities, we measure the alignment between the feature spaces of LLMs and pretrained vision models, following the methodology of the Platonic Representation Hypothesis (Huh et al., 2024a). For this analysis, we use image-caption pairs from the Wikipedia-based Image Text (WIT) dataset (Srinivasan et al., 2021). Given an image $x_i$ and its caption $y_i$, we compute vision and language kernels, $K_{\text{vision}}$ and $K_{\text{lang}}$, from their respective model representations, $f_{\text{vision}}$ and $f_{\text{lang}}$:

$$K_{\text{vision}}(i, j) = \langle f_{\text{vision}}(x_i), f_{\text{vision}}(x_j) \rangle$$
$$K_{\text{lang}}(i, j) = \langle f_{\text{lang}}(y_i), f_{\text{lang}}(y_j) \rangle$$

We then assess the alignment between these kernels using the mutual nearest-neighbor ($m_{NN}$) metric, which calculates the average overlap of the $k$-nearest neighbor sets (with $k = 20$) for each pair. The final alignment score for a given LLM is reported as the average of its $m_{NN}$ scores against three strong vision backbones: ViT-Large (Dosovitskiy et al., 2021) (trained on ImageNet-21K (Deng et al., 2009)), DINOv2-Giant (Oquab et al., 2023), and CLIP-Huge (Radford et al., 2021).

For all evaluations, we fix the random seed as 42 and use temperature 0 for testing to get consistent results.

## C  RELATED WORK

### C.1  FROM LLMs TO MLLMs.

With the rapid development of LLMs (Radford et al., 2021; Google, 2023; Touvron et al., 2023), a direction of work extends LLMs to Multimodal LLMs. Pioneering works like Flamingo (Alayrac et al., 2022) and BLIP-2 (Li et al., 2023) connected pre-trained vision encoders to LLMs using connectors like cross-attention modules. Later models such as LLaVA (Liu et al., 2023a) demonstrate that even with a projection layer, LLMs can be extended MLLM with visual instruction tuning. This adapter-style architecture has been widely explored in numerous subsequent works (Liu et al., 2024a; Tong et al., 2024a; Laurençon et al., 2024; Liu et al., 2024b; AI@Meta, 2024; Bai et al., 2025b; Zhu et al., 2025; Team et al., 2025; Lin et al., 2024).

The success of visual instruction tuning has enabled open-source multimodal models to achieve performance even comparable to proprietary counterparts (Google, 2023; xAI, 2024; OpenAI, 2024). This success underscores that multimodal capabilities in adapted LLMs largely emerge through instruction tuning, effectively unlocking knowledge already embedded in pretrained language models (Zhou et al., 2024). Furthermore, recent studies (Tong et al., 2024a; Laurençon et al., 2024) highlight that improvements in the underlying language model remain the most impactful means to improve multimodal performance. Inspired by these insights, our work investigates the visual priors and inherent multimodal potential embedded within pretrained LLMs.

Though there are different ways of connecting vision to LLMs like the use of discrete tokenization (Wang et al., 2024; Deng et al., 2025; Team, 2024), we focus on adapter-style architectures, which are most widely used and permit clean analysis of visual priors from language pre-training.

### C.2  THE ROLE OF DATA IN SHAPING FOUNDATION MODEL CAPABILITIES.

Pretrained LLMs encode rich latent knowledge—even across modalities—depending heavily on the nature of their training data (Kaplan et al., 2020; Grattafiori et al., 2024; Han et al., 2022; Rae et al., 2021; Penedo et al., 2023; Lu et al., 2022a; Mayilvahanan et al., 2025). This has shifted the focus of research from simply scaling data to understanding the role of data and then strategically curating it to unlock specific, powerful abilities (Allen-Zhu, 2024; Aryabumi et al., 2024; Ye et al., 2024; Shinnick et al., 2025).

A prominent example, and one highly relevant to our findings, is the strategic inclusion of reasoning-centric data like `code`. Research has consistently shown that pretraining on a mix of text and `code` does more than just improve coding skills; it significantly enhances a model's foundational reasoning and ability to understand abstract, structural patterns (Muennighoff et al., 2023; Aryabumi et al., 2024; Ma et al., 2023; Zhang et al., 2025). This suggests that the pre-training data mixture may endows the model with latent, generalizable structures that can be activated for tasks beyond their original domain.

This has established a central challenge in the field: determining the optimal data mixture to cultivate these desired foundational abilities (Chen et al., 2024; Ma et al., 2023; Xie et al., 2023; Touvron et al., 2023; Grattafiori et al., 2024; Zhang et al., 2024; Held et al., 2025; Bai et al., 2024; Albalak et al., 2023; Shukor et al., 2025a). This has spurred a move beyond simple heuristics toward quantitative frameworks which aim to predict a model's performance based on different data blends, thereby guiding the search for an optimal mixture.

However, much of this prior work has focused on optimizing data mixtures for core language proficiency. As we transition from LLMs to MLLMs, a critical question emerges: how do these text-only pre-training choices influence the model's visual priors and the potential for multimodal capabilities? Our work directly addresses this gap. We extend the investigation of language data composition's impact from the purely linguistic to the visual domain, systematically analyzing how different text sources contribute to the emergent visual priors in LLMs and seeking data mixture to help them "learn to see" more effectively from text pre-training.

## D  Limitations and Future Research Directions

While this work provides a systematic analysis of visual priors in LLMs, it is subject to several limitations that open avenues for future research.

First, our investigation primarily centers on adapter-style MLLM architectures. While this is a prevalent and effective paradigm, our findings may not fully generalize to other approaches, such as those that employ discrete visual tokenization (Team, 2024; Wang et al., 2024; Deng et al., 2025; Wu et al., 2024) or involve end-to-end joint training of vision and language components (Diao et al., 2024; Tao et al., 2025; Diao et al., 2025; Shukor et al., 2025b). In these latter cases, language and vision data are co-trained, making it hard to identify the priors originating solely from language. The dynamics of how visual priors are formed and utilized could differ in these models, which leaves a promising future direction.

Second, a significant area our study does not address is the safety and ethical implications of these learned visual priors. Language corpora are known to contain societal biases, stereotypes, and potentially harmful content (Bengio et al., 2024; Qu et al., 2023). Our analysis focused on capability, but did not investigate whether these text-based priors encode biased visual associations (*e.g.*, linking certain objects or roles to specific genders or races) that could manifest as harmful generation or classification behavior in a downstream MLLM. A thorough audit of the fairness and safety of these emergent priors is a critical next step.

Finally, our study is confined to the domain of static images, leaving the exploration of visual priors for dynamic modalities, such as video understanding, as an open question. For example, the temporal knowledge important for video understanding might be learned more from story-related data like `literature`. Investigating how different textual sources contribute to priors for temporal reasoning, action recognition, and causality in video is a rich area for future work.

## E  Broader Impact

Our research provides a systematic analysis of how prior visual capabilities emerge in LLMs from language-only pre-training, shifting the paradigm from accidental discovery to deliberate cultivation. While our work focuses on capability, the textual data used for pre-training contains societal biases. A significant risk is that these models could learn and reinforce harmful visual stereotypes, which could then manifest in downstream multimodal systems.

Nevertheless, our findings primarily help researchers and developers understand the nature and origins of these visual priors. We demonstrate that these priors are not a single, uniform block but are composed of separable perception and reasoning components, each cultivated by different types of text.

This deeper understanding provides a clear, actionable path for more efficiently cultivating these abilities. Instead of relying on serendipity scaling, teams can now strategically curate their text-only pre-training data to deliberately build a stronger foundation for vision tasks before multimodal training even begins. This targeted approach not only improves the final model's multimodal performance but also reduces the computational resources required, offering a more sustainable methodology for creating the next generation of vision-language models.

## F  Visual World and Reasoning-centric Language Data Classification

This section provides the detailed classification setting and results for the visual world and reasoning-centric pre-training data sources discussed in the main paper. We use a 32B dense LLM (Yang et al., 2025a) to perform a multi-label classification on 1024-token segments from each data source. Below is the full prompt provided to the LLM for the visual world and reasoning-centri data classification task. The prompt instructs the model to perform a multi-label classification on text segments, assigning one or more predefined categories that describe the content.

| Data sources | Reasoning Categories (%) | | | | Visual Categories (%) | | | |
|---|---|---|---|---|---|---|---|---|
| | code reasoning | math reasoning | science reasoning | reasoning combination | visual concept | visual attribute | visual relationship | visual combination |
| web-crawl | 3.5 | 3.6 | 8.6 | 10.0 | 27.7 | 14.1 | 13.5 | 26.9 |
| encyclopedia | 0.5 | 2.3 | 3.0 | 3.4 | 12.2 | 2.9 | 6.0 | 12.4 |
| academia | 21.0 | 68.2 | 74.4 | 83.3 | 5.2 | 0.7 | 5.1 | 5.3 |
| literature | 0.3 | 1.3 | 8.9 | 9.1 | 33.4 | 8.4 | 27.9 | 33.5 |
| math | 31.1 | 81.7 | 83.2 | 92.9 | 7.8 | 2.8 | 6.6 | 8.0 |
| code | 96.7 | 13.3 | 6.8 | 97.3 | 3.3 | 2.0 | 2.4 | 3.7 |

Table 4: **Conceptual categories of key pre-training data corpus (%).** The table shows the percentage of text segments from each data corpus classified into one of the conceptual categories.

---

**Prompt for LLM-based Visual World and Reasoning-centric Data Classification**

Analyze the provided text paragraph. Classify its content by identifying the primary concepts and domains using only the categories listed below. Select categories that represent the text's significant content.

- `visual concept`: Language for naming visual entities (e.g., objects, people, places, actions, scenes).

- `visual attribute`: Language describing visual properties (e.g., color, size, shape, texture, style).

- `visual relationship`: Language describing spatial or part-whole relations between entities.

- `code reasoning`: Content centered on algorithmic problem-solving, logical coding implementation, and software engineering challenges.

- `math reasoning`: Content focused on logical math reasoning, proof construction, and the application of mathematical principles to solve problems.

- `science reasoning`: Focuses on scientific reasoning, including hypothesis testing, data analysis, and modeling of complex systems.

If none of the above categories apply, output None.

---

The percentages in Table 4 represent the proportion of text segments within each data source that were assigned a given label.

## G   Visual Instruction Tuning Data Classification

We use a 7B VLM (Yang et al., 2025a) to perform a multi-label classification on the Cambrian-7M data. We prompt the model to classify visual instruction tuning data into three categories: perception-oriented, reasoning-oriented and neither.

---

**Prompt for VLM-based Perception and Reasoning-centric Data Classification**

Please classify the following into three categories: OTHERS, PERCEPTION, or REASONING. OTHERS covers most standard questions including visual analysis, general knowledge, and basic inquiries. PERCEPTION is for questions that require visual speculation or interpretation. REASONING is for complex analytical tasks that require advanced theoretical frameworks or sophisticated analysis. When classifying, consider what type of cognitive process would be most relevant to answering the question effectively.

---

## H   Robust Parsing for VQA Evaluations

A significant challenge in the automated evaluation of VQA is that models often generate conversational or free-form text instead of a single-letter answer. A naive parsing strategy that only checks for

an exact match to the ground-truth letter (e.g., "B") would unfairly penalize models that provide a correct but differently formatted response.

To illustrate, consider a simple VQA task:

- Question: "What is the primary object in the image?"
- Options: `(A) A bicycle`, `(B) A car`, `(C) A tree`
- Ground Truth: `B`

A model could correctly answer in multiple ways, such as *"The answer is (B)"*, *"A car"*, or *"The image shows a car"*. To capture all these valid responses, our evaluation protocol employs a robust, hierarchical parsing strategy. The logic is executed as a sequence of prioritized steps, stopping as soon as a valid answer is found:

1. **Explicit letter extraction**: The parser first searches for high-confidence patterns that directly indicate the chosen option letter. It uses regular expressions to find formats like:
   - *"The correct answer is (B)"*
   - *"Answer: B"*
   - Outputs starting or ending with *'(B)'*
   - An output that is simply *'B'* or *'B.'*

2. **Exact option text matching**: If the first step fails, the parser extracts the text associated with each option from the prompt (e.g., "A bicycle", "A car", "A tree"). It then checks if the model's generated text is an *exact, case-insensitive match* for any of these option strings.
   - Example caught: A model output of *"A car"* would be correctly mapped to option `B`.

3. **Substring matching**: As a final fallback, the parser checks if the text of any option appears as a substring within the model's generated output. This handles more verbose, conversational answers.
   - Example caught: A model output of *"The image features a car driving down the street"* would be correctly mapped to option `B` because "a car" is present.
   - To prevent ambiguity (e.g., if one option was "car" and another was "race car"), this step returns the longest matching option text found in the response.

This multi-tiered parsing strategy ensures a comprehensive and fair evaluation across all models, regardless of their verbosity or adherence to specific formatting instructions. It allows us to more accurately measure the model's underlying visual capabilities rather than its ability to follow formatting rules.

## I  MULTI-LEVEL EXISTENCE BENCHMARK CONSTRUCTION

This section describes how we constructed our benchmark using publicly available SA-1B and ADE20K datasets. We selected images with ground-truth segmentation masks and calculated the proportion of the image area each object occupied. Based on this, we created three splits: 0–30 for small objects, 30–60 for medium, and 60–100 for large, dominant objects. For each image, we created a multiple-choice query to test object existence, sampling distractors from the dataset vocabulary and filtering them to exclude objects present in the ground truth. As this process is open-vocabulary, we use an LLM to filter out distractors that correspond to objects already present in the image but under different names. This ensures the distractors remain plausible but incorrect, providing a granular evaluation of a model's ability to identify objects across a wide range of sizes.

## J  MULTI-LEVEL EXISTENCE BENCHMARK RESULTS

serves as a crucial tool for pushing models toward more universal visual understanding. This section presents the performance of top models on our Multi-Level Existence (MLE) benchmark, which evaluates their ability to identify objects of varying sizes. The results, detailed in Table 5, reveal

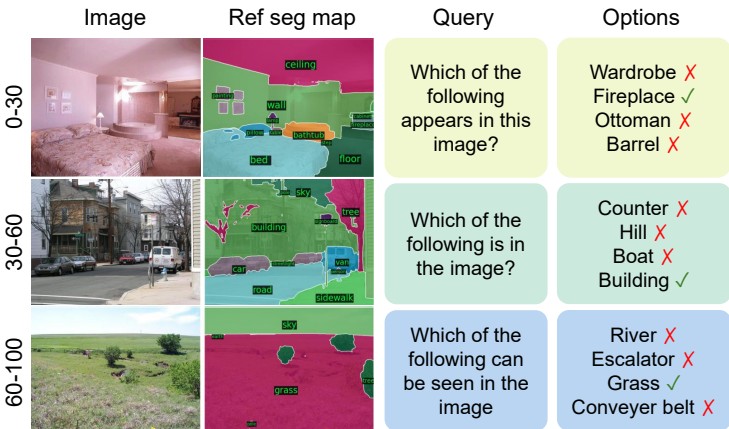

Figure 12: **MLE Benchmark Examples.** The figure provides examples from the MLE-Bench, illustrating how the dataset is partitioned based on the ground-truth object size from reference segmentation maps. For instance, in the 0-30 split, the target object (a fireplace) constitutes a small fraction of the image. In contrast, the 60-90 split features a correct object (grass) that covers a substantial portion of the image.

| Model | 0–30 | 30–60 | 60–100 | Overall |
|---|---|---|---|---|
| gpt-5 | 73.63 | 90.69 | 86.31 | 82.97 |
| gpt-4o-mini | 68.72 | 85.10 | 87.94 | 79.32 |
| gemini-2.5-flash | 58.88 | 80.66 | 80.74 | 72.11 |
| claude-opus-4-20250514 | 61.20 | 76.36 | 59.86 | 66.58 |

Table 5: **Model performance on the MLE-Bench.** Results are reported in three splits based on object size in percentage (0-30, 30-60, 60-100), along with a weighted overall accuracy, evaluating the ability of different models to identify objects of varying sizes.

distinct performance profiles and highlight that robust perception across all object scales remains a challenge.

gpt-5 achieves the highest overall performance with a weighted accuracy of 82.97%, demonstrating strong capabilities across all categories. It particularly excels at identifying medium-sized objects (90.69%), which form the largest portion of the visual scene, while still maintaining competent, albeit lower, performance on small objects (73.6%). In contrast, other models exhibit more pronounced trade-offs. gemini-2.5-flash struggles significantly with small objects, scoring only 58.88%, which is over 20 percentage points lower than its performance on medium and large objects. This indicates a potential weakness in fine-grained perception. Similarly, claude-opus-4-20250514 shows a notable drop in performance on large, dominant objects (59.86%) compared to smaller ones.

These findings underscore the utility of the MLE benchmark in diagnosing model weaknesses. While some models demonstrate strong general performance, universal recognition across different scales is not guaranteed. The benchmark serves as a tool for driving progress toward more comprehensive and reliable visual understanding.

## K    BLIND VISUAL INSTRUCTION TUNING

We also introduce a "blind visual instruction tuning" trick that provides a more effective starting point for visual adaptation. This trick involves an initial instruction tuning phase using only the textual data while withholding the corresponding images. This initial "blind" stage allows the model to first focus on learning the instruction-following format of the task. Consequently, the subsequent standard tuning phase with images can be more dedicated to learning the core vision capabilities, rather than simultaneously learning how to follow instructions. Furthermore, this process enables the model to effectively leverage its pre-existing language priors to solve VQA questions that may not strictly

| Model | Language | | Vision | | | | |
|---|---|---|---|---|---|---|---|
| | ppl | avg acc | General | Knowledge | OCR&Chart QA | Vision-Centric | Overall |
| Lang-Favorable | 8.72 | 0.647 | 46.92 | 28.35 | 21.49 | 46.31 | 37.32 |
| Lang-Favorable (+Blind) | | | 48.16 | 30.30 | 20.77 | 47.01 | 38.20 |
| Balanced | 7.49 | 0.655 | 49.59 | 29.02 | 23.63 | 46.59 | 38.64 |
| Balanced (+Blind) | | | 50.90 | 31.25 | 22.60 | 47.32 | 39.56 |

Table 6: **Performance comparisons of the Lang-Favorable and Balanced models across both language and vision-language benchmarks with blind visual instruction tuning trick.** The table summarizes key language metrics (perplexity and accuracy) and provides average scores for a suite of vision tasks, categorized as General, Knowledge, OCR & Chart QA, and Vision-Centric. It also shows the impact of applying our blind visual instruction tuning trick (+Blind). The blind tuning method provides an additional performance boost for both models.

require visual input, a known phenomenon and potential "shortcut" on some benchmarks (Tong et al., 2024a). This trick can lead to broad performance improvements in most of the tasks.

This trick should not be a standard in practice, since when no images are provided, models should identify their absence rather than encouraging more hallucination. This type of hallucination may persist even in frontier models including GPT-5 thinking, Gemini 2.5 Pro, and Claude Opus 4.1. Models will exhibit such hallucinatory behaviors when answering VQA questions without actual visual context given. Some examples are presented in Appendix L. This highlights a systemic issue that warrants more future investigation. We leave this as an optional trick and introduce this phenomenon to the community for future investigation. This also shows that "vision" can be "learned" from the language side through an unconventional, shortcut-based mechanism.

We apply this trick to Language-favorable model and Balanced model. The trick yields additional overall performance gains for both models by clear margins, as shown in Table 6. The performance gains are most pronounced in the Knowledge categories, while tasks in OCR & Chart VQA, conversely, suffer a performance drop. This pattern suggests that while the blind tuning phase provides a more effective initialization for the subsequent visual instruction tuning, it gains more from strengthening the model's ability to leverage its internal knowledge and reasoning priors to "shortcut" the problem. This is highly beneficial for knowledge-intensive tasks where answers can be inferred or reasoned from textual context and the LLM's pre-existing knowledge.

## L  HALLUCINATIONS IN BLIND VQA

To test for hallucinations, we present two randomly selected questions (with slight rephrasing to increase diversity) from our MLE-Bench to various models without the corresponding images. The examples below are from models also chosen randomly, primarily via LMArena (Chiang et al., 2024). We observe that stronger models do not necessarily guarantee fewer hallucinations in this blind VQA setting.

Question 1

**Question:** Which of the following is visible in the image: Choose from: (A) purple and white striped wall, (B) a person's arm, (C) a purple triangle, (D) building. Please answer with a single phrase.

- - - - - - - - - - - - - - - - - - - - - - - - - - - - - - - - - - - - - - - - - - - - - - -

**Answers:**
**gemini 2.5 pro:**
A purple and white striped wall

**gpt-5 thinking:**
a person's arm

**qwen-max-2025-08-15:**
Red and white striped wall

**o4-mini-2025-04-16:**
a person's arm

**mistral-medium-2508:**
(D) building

**gemma 3 27B:**
a person's arm.

**o4-mini-2025-04-16:**
a person's arm

**kimi-k2-0711-preview:**
(B) a person's arm

**qwq-32b:**
(C) a purple triangle

**gpt-oss-120b:**
I'm sorry, but I can't see images. Could you describe what's in the picture?

**kimi-k2-0711-preview:**
(B) a person's arm

**mistral-medium-2508:**
(D) building

**llama-4-scout-17b-16e-instruct:**
B

**o3-2025-04-16:**
building

**claude-sonnet-4-20250514:**
I don't see any image attached to your message. Could you please share the image you'd like me to analyze?

**phantom-0807-1:**
(D) building

**claude-opus-4-1-20250805:**
(B) a person's arm

---

**Question 2**

**Question:** What can you see in the image: Choose one from: (A) a blue jersey, (B) the street, (C) araff logo, (D) a blue and white computer screen. Please answer directly.

---

**Answers:**
**mistral-medium-2508:**
D) a blue and white computer screen.

**qwen3-235b-a22b-instruct-2507:**
(C) araff logo

**qwen3-235b-a22b:**
(A) a blue jersey

**amazon.nova-pro-v1:0:**
(A) a blue jersey

**phantom-0807-3:**
Without access to the image, it is impossible to determine the correct answer definitively. However, based on the options provided, D) a blue and white computer screen is a common element in images and could be a plausible choice.

**mistral-medium-2508:**
(D) building

**gpt-oss-20b:**
I'm sorry, but I can't see the image.

**qwen3-30b-a3b-instruct-2507:**
(D) a blue and white computer screen.

**gpt-4.1-2025-04-14:**
(D) a blue and white computer screen.

**phantom-0807-1:**
(C) araff logo

**claude-sonnet-4-20250514-thinking-32k:**
I don't see any image attached to your message. Could you please share the image you'd like me to analyze? Once you upload it, I'll be happy to choose from the options you've provided.

**gpt-5-chat:**
(D) a blue and white computer screen.

**mistral-medium-2508:**
(D) building

**claude-opus-4-1-20250805:**
Looking at the image, I can see (B) the street. The image shows an urban street scene with buildings, trees, and what appears to be a road or pathway.

**folsom-0811-1:**
I cannot view or analyze images directly. However, if you describe the image or provide more context, I can help determine the correct answer. Since I must choose one option blindly, I'll note that (D) a blue and white computer screen is a common object that might align with typical visual descriptions, but this is a guess. For accuracy, please describe the image!

## M ADDITIONAL QUALITATIVE EXAMPLES

To illustrate the differences between our models, we provide additional chat examples. These include one from `reasoning combination` (Figure 13) and one from `visual combination` (Figure 14). Each figure shows five different proportions of pre-training data.

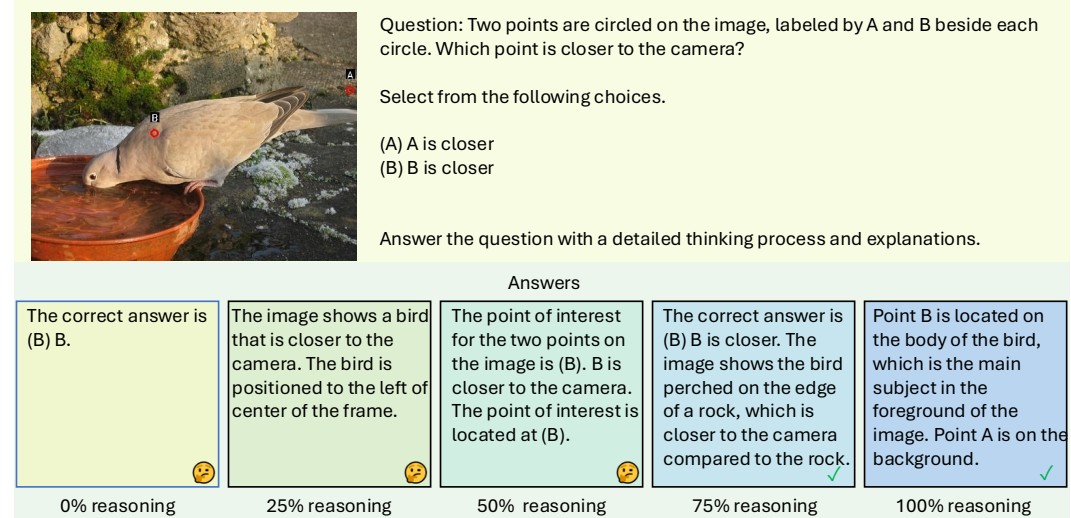

Figure 13: **Qualitative impact of reasoning-centric data on visual spatial reasoning.** The figure shows answers from five models—pre-trained with 0% to 100% reasoning combination data—to a visual question requiring depth perception. The answers demonstrate a clear improvement in reasoning quality: the model with 0% reasoning data gives a blunt answer, while the model with 100% reasoning data provides a detailed explanation correctly applying concepts of foreground and background.

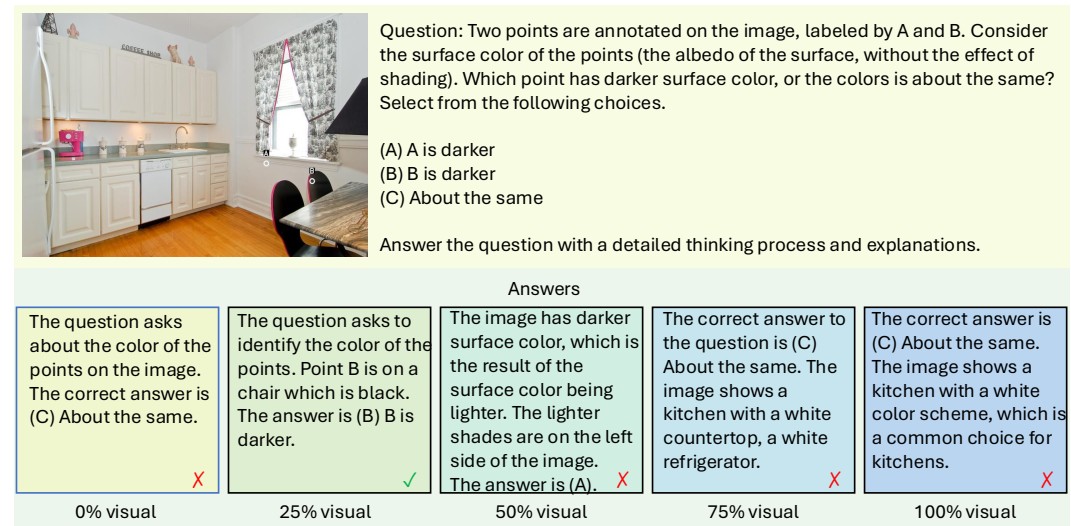

Figure 14: **Qualitative impact of visual world data on complex visual perception.** The figure displays answers from five models—pre-trained with 0% to 100% visual combination data—to a question requiring an understanding of color constancy. The results show that while the model with 25% visual data provides the correct answer with reasoning relevant to the core visual principle, models trained on more visual data offer incorrect answers and flawed explanations. This suggests that simply increasing descriptive visual text does not necessarily cultivate a deeper perceptual understanding.

