# OpenReview forum: "Learning to See Before Seeing: Demystifying LLM Visual Priors from Language Pre-training"
_ICLR.cc/2026/Conference — ICLR 2026 Oral_

### Official Review · Reviewer_JSpZ · 2025-10-31

**Soundness:** 4
**Presentation:** 3
**Contribution:** 4
**Rating:** 8
**Confidence:** 3

**Summary:**

This paper presents a comprehensive study on how the vision-related capabilities of MLLMs—including visual perception and reasoning—emerge during text-only pretraining. To investigate this, the authors conduct extensive controlled experiments by training LLMs of different sizes under various data mixture settings, carefully varying the proportion of image-caption pairs of different types.

These experiments yield a set of valuable insights, revealing where the model’s perception and reasoning abilities originate from even in the absence of real visual input. The results highlight the importance of descriptive language data and reasoning data in shaping vision-related capabilities, and identify the optimal data mixture ratios for strong downstream VQA performance.

Leveraging these findings, the authors scale up the training to approximately 1 trillion tokens and train a larger model following the recommended data composition. The improved performance of this scaled model further validates the effectiveness of the insights derived from the ablation studies.

**Strengths:**

* The problem investigated in this paper is highly important and relevant to the MLLM community, yet has been largely overlooked in prior work. Understanding how vision-related capabilities emerge from text-only pretraining provides valuable insight into the foundations of multimodal learning.

* The conclusions drawn from the extensive experiments are both instructive and insightful. They offer practical guidance for future model development, particularly in designing effective data mixtures and scaling strategies for training more capable MLLMs.

* The paper is well-organized, clearly written, and easy to follow. The presentation is informative, with a logical flow that effectively conveys the key ideas and findings.

**Weaknesses:**

* All VQA questions are divided into several categories, such as General VQA, Knowledge-based VQA, and OCR & Chart VQA. However, these category definitions may appear abstract to readers, especially without concrete examples. Including one or two representative examples for each type in the main text or a table would significantly improve clarity and help readers better understand the distinctions between the tasks.

* I understand that the 9-page limit constrains the space available for presentation. Nevertheless, some conclusions in the supplementary materials also contain valuable insights that support the paper’s findings. While they may not fit in full, briefly referring to them in the main text would help draw readers’ attention to these contents and enhance the overall impact of the work.

**Questions:**

See Weaknesses.

---

> ### Author Response · Authors · 2025-11-19
> **Thank you & reply**
>
> Thank you for your positive evaluation and careful reading of our manuscript, including the materials in the Appendix that were not mentioned in the main paper! We appreciate your thoughtful questions, which will help us improve the paper's clarity and presentation.
>
> **W1: Including one or two representative examples for each VQA type.**
>
> This is an excellent suggestion. We have created a new figure to present representative examples for each VQA type and have placed it in the evaluation protocol (page 3).
>
> **W2: Briefly referring to conclusions in the supplementary materials that also contain valuable insights.**
>
> Following your suggestion, we have added references to key experiments and conclusions from the Appendix to the main paper. In addition to supplementary experimental designs, settings, and more visualization examples, our Appendix includes the following:
>
> (1) Three hypotheses with experiments on the structure of the perception prior, the universality of reasoning skills, and the representation alignment with respect to language data structure.
>
> (2) The results of several MLLMs on the proposed MLE-bench.
>
> (3) A technique we call "blind visual instruction tuning" and the related phenomenon of blind hallucinations in VQA.
>
> We have added short conclusions about all three hypotheses in points (1), as well as short previews about points (2) and (3) in suitable positions in the main paper with references to the corresponding sections of the Appendix (page 8,9, and 10). The revision part is highlighted in blue color.

---

> > ### Comment · Reviewer_JSpZ · 2025-11-20
> >
> > Thanks for the responses. My questions are mostly addressed and I think this paper is currently in a good shape for acceptance.

---

> > > ### Author Response · Authors · 2025-11-24
> > >
> > > Thank you very much for the quick follow-up. We are very glad to hear that your concerns have been addressed, and we thank you again for your constructive comments!!

---

### Official Review · Reviewer_1T9p · 2025-10-31

**Soundness:** 3
**Presentation:** 4
**Contribution:** 3
**Rating:** 8
**Confidence:** 5

**Summary:**

Based on the common practice of finetuning Multimodal Large Language Models (MLLMs) from pretrained Large Language Models (LLMs), this paper analyzes how LLM pretraining influences downstream MLLM finetuning performance. It focus primarily on how LLM pretraining data construction affects the downstream MLLM performance. The results reveal several interesting findings including aspects of
- Scaling law of LLM pretraining on MLLM finetuning
- Source of visual prior
- Data arrangements
- Structure of visual priors

Guided by these findings, this paper pretrains a new LLM using a reorganized language dataset, achieving improved downstream performance for MLLM finetuning.

**Strengths:**

- Well-motivated problem with clear relevance to current MLLM construction practices.
- Sufficient evaluation breadth and depth, with extensive controlled studies across data mixtures and scales.
- In-depth findings that are coherent with the presented experiments and lead to a concrete, data-centric pretraining recipe.

**Weaknesses:**

- Overclaim in the abstract that "perform visual tasks without ever having seen an image". The paper does not actually demonstrate an MLLM that performs visual tasks without any visual training.
- Unclear vision-input processing in the default settings. It is not clear whether the default setup uses any vision encoders or just a MLP adapter. And how to process the vision inputs.
- Since stronger LLMs typically yield stronger MLLMs after adaptation, it remains ambiguous how much of the reported gains come from claimed "visual priors" versus general language capacity. The paper only reports perfomance on vision tasks, but lacks evaluation of the language capacity of each mdoel. So it is not very clear whether it is really benefits from visual prior or just stronger langauge capacities.

**Questions:**

How is model trained with "0B" data obtained in Figure 1? Why a model with no training data can also get non-trivial performance?

---

> ### Author Response · Authors · 2025-11-19
> **Thank you & reply**
>
> We sincerely thank you for the constructive assessment and for raising several important questions that will help us improve the paper's clarity and precision! We are deeply encouraged by your positive rating. Here are our responses to all your concerns and questions.
>
> **W1: perform visual tasks without ever having seen an image.**
>
> We acknowledge that our original phrasing, "perform visual tasks without ever having seen an image," could be interpreted too broadly. Our intention was to highlight the phenomenon of programmatic visual knowledge (lines 47-52), where LLMs can generate executable code to render complex 2D and 3D scenes without any image-based training, a capability supported by prior work [1,2].
>
> To be more precise, we have revised the sentence in the abstract to: "perform symbolic visual generation tasks without ever having seen an image."
>
> [1]: A vision check-up for language models. CVPR 2024.
>
> [2]: 3D-GPT: Procedural 3D Modeling with Large Language Models, 3DV 2025.
>
> **W2: vision-input processing.**
>
> Our MLLM adaptation follows a standard and widely-used LLaVA-like architecture. Specifically, we use a frozen vision encoder (MetaCLIP-B/16 by default) to extract image features. These features are then mapped into the LLM's word embedding space via a trainable 2-layer MLP projector.
>
> We also provide a more detailed explanation of this setup in Appendix B.1 (MLLM adaptation setting in page 21).
>
>
> **W3:  visual priors versus general language capacity.**
>
> This is an excellent and insightful point! At large scales, stronger general language capacity often correlates with better multimodal performance, as these models are massively trained with a wide variety of pre-training data. Our aim was to use strictly controlled experiments to isolate the specific impact of data composition on visual priors, rather than on general language capacity. Our findings reveal that this correlation between language and vision does not always hold in such controlled settings. We describe two groups of experiments to demonstrate this.
>
> **Contradictory trends in data mixture search (Finding 4, Table 2):** When searching for a "vision-aware" mixture, we observed no direct positive correlation between performance on language and vision benchmarks. The optimal mixture for vision (mix10) was not the best for language, and conversely, the optimal mixture for language (mix0) was not the best for vision.
>
> **Single-source pre-training (Finding 2, Figure 2):** Our experiments training 16 models on single data sources provide further evidence. The model trained exclusively on code achieved the best overall vision performance. However, its general language capacity was severely limited due to the lack of natural language exposure. Below, we present the language performance results for these 16 models alongside their corresponding average VQA performance.
>
> | Source | text_acc | vqa_acc |
> | :--- | :---: | :---: |
> | code | 44.2 | 32.1 |
> | arts | 47.2 | 32.0 |
> | food | 49.4 | 31.5 |
> | web-crawl | 52.8 | 31.3 |
> | math | 50.0 | 31.1 |
> | academia | 42.3 | 31.1 |
> | philosophy | 45.9 | 30.7 |
> | law | 43.8 | 30.7 |
> | economics | 45.1 | 30.7 |
> | politics | 44.8 | 30.5 |
> | literature | 48.8 | 30.5 |
> | computer sci | 45.1 | 30.3 |
> | encyclopedia | 45.3 | 30.2 |
> | q-a forum | 43.6 | 30.1 |
> | medicine | 48.6 | 29.9 |
> | biology | 47.0 | 29.8 |
>
> The Spearman’s correlation between text_acc and vqa_acc is only 0.149. Notably, the model trained on code, which achieves the best VQA performance, has a very low text accuracy (rank 13/16). These results reinforce that the correlation between language and vision does not always hold in controlled settings and demonstrate that our experimental design effectively isolates the impact on visual priors over general language capacity.
>
> **Q1:  How is model trained with "0B" data obtained in Figure 1?**
>
> The "0B data" model in Figure 1 is a randomly initialized LLM. It has undergone zero language pre-training. This model serves as a baseline to measure the capabilities gained from the subsequent adaptation stages.
>
>
> **Q2: Why a model with no training data can also get non-trivial performance?**
>
> The model's non-trivial performance, despite no pre-training, is because it still undergoes the two standard adaptation phases: 1) visual alignment and 2) language+vision instruction fine-tuning. These stages provide it with some knowledge to perform very basic VQA tasks. Furthermore, our robust VQA parsing strategy (detailed in Appendix H, page 25-26) is designed to extract correct answers even when a model fails to follow instructions perfectly, which is common for weaker models like the 0B baseline. This ensures we are still measuring its capability, leading to its non-trivial score.

---

### Official Review · Reviewer_fgou · 2025-11-01

**Soundness:** 3
**Presentation:** 3
**Contribution:** 3
**Rating:** 6
**Confidence:** 2

**Summary:**

## Summary

This paper is a pioneering work that uses a proxy visual task framework (P-SET) to quantify and dissect the **visual priors** that emerge in Large Language Models (LLMs) from **text-only** pre-training, decomposing them into separable perception and reasoning components. The study reveals that these priors follow unique scaling laws, where the reasoning prior scales poorly, providing the MLLM community with critical **descriptive insights** to guide future model alignment and pre-training strategies.

**Strengths:**

## Strengths

1.  **S1: Pioneering Dissection of Emergent Visual Knowledge.**
    This work represents a **pioneering effort** to systematically dissect and analyze the implicit visual knowledge (priors) that emerges purely from **text-only** pre-training in Large Language Models. By demonstrating the existence and growth of these priors, the paper provides a crucial, non-trivial insight into the mechanisms by which LLMs acquire world knowledge, laying the foundation for **mechanistic interpretability** studies in the multimodal domain.

2.  **S2: Novel Framework for Operationalizing and Separating Priors.**
    The paper introduces a **novel and comprehensive framework** using proxy visual tasks (perception vs. reasoning) to quantitatively *operationalize* and *separate* the distinct components of the visual prior. This methodology allows for the first time a clear, quantitative distinction between different types of emergent visual capabilities, moving beyond anecdotal evidence to **systematic, scalable measurement**.

3.  **S3: Key Scaling and Developmental Insights for MLLM Construction.**
    The work yields **critical, quantitative insights** into how the perception and reasoning priors scale with model size. The key finding that the **reasoning prior scales poorly** compared to the perception prior provides a valuable, data-driven hypothesis for MLLM researchers, guiding future efforts to strategically adjust data composition or architecture to unlock superior visual reasoning capabilities.

4.  **S4: Rigorous Construction of Proxy Visual Tasks (P-SET).**
    The paper's construction of the P-SET benchmark, specifically designed to isolate and measure perception and reasoning capabilities within the text modality, is highly rigorous. This set of tasks effectively serves as a crucial tool for **probing** the latent knowledge structure of LLMs, and the benchmark itself is a valuable public contribution to the field of MLLM analysis.

5.  **S5: Strong Empirical Evidence via Causal Intervention (Minimal Fine-Tuning).**
    The central hypothesis—that these priors exist and are beneficial—is supported by strong empirical evidence showing that LLMs with higher measured priors achieve better performance in MLLM tasks (e.g., VQA) after **minimal multimodal fine-tuning**. This suggests that the latent knowledge is indeed functional and ready to be "unlocked," validating the study's relevance to the efficiency of MLLM alignment.

**Weaknesses:**

## Weaknesses

1.  **W1: Lack of Mechanistic Evidence for the Separation of Priors.**
    The central claim is the separation of visual priors into "perception" and "reasoning" components. This distinction relies heavily on an **operational definition** (performance on specific proxy tasks) rather than **mechanistic evidence**.
    * The authors **fail to demonstrate** that these priors are truly **orthogonal or separable** in the LLM's internal representations (e.g., via representation similarity analysis, or targeted causal interventions on specific latent dimensions).
    * This lack of a solid **mechanistic foundation** suggests that the claimed separation might be an artifact of the task design itself rather than an inherent property of the emergent knowledge within the LLM.

2.  **W2: Fatal Causal Confounding of Textual Knowledge and Visual Priors.**
    The core, high-impact thesis is that visual priors emerge from **text-only** pre-training. However, the pre-training data contains vast amounts of **structured, descriptive visual language** (e.g., "the red square is to the left of the blue circle").
    * The paper fails to **rigorously rule out** that the LLM is merely encoding **linguistic knowledge about visual relationships** (i.e., the language semantics of relational phrases) rather than truly emergent **visual spatial knowledge**.
    * This **causal confounding** between linguistic encoding and visual priors significantly weakens the claim that LLMs learn to "see" before seeing an image, as the identified "prior" could simply be complex language fluency.

3.  **W3: Lack of Prescriptive Guidance for MLLM Design (The Analysis-Synthesis Gap).**
    The work is primarily **descriptive**, successfully identifying the existence and scaling trends of these priors (e.g., reasoning prior scales poorly). However, it fails to offer **actionable, prescriptive guidance**.
    * For instance, given the finding that the reasoning prior scales poorly, can the authors propose a **concrete, quantifiable text pre-training data optimization strategy** (e.g., adjusting the ratio of relational reasoning text) and empirically prove that this modification **more efficiently** boosts the reasoning capability?
    * This lack of **analysis-to-synthesis** guidance limits the practical value of this research for constructing or optimizing the pre-training stage of next-generation MLLMs.

**Questions:**

## Questions

1.  **Q1: Mechanistic Validation of Prior "Separability."**
    The core claim is the operational separation of visual priors (perception vs. reasoning) based on P-SET tasks. Can the authors provide evidence at the level of the LLM’s internal representations (e.g., via Representation Similarity Analysis, CCA, or causal intervention) to prove that these priors are **orthogonal or uncorrelated** in the latent space? Lacking this mechanistic evidence, how do we rule out that the separation is merely an artifact of the **proxy task design** (P-SET)?

2.  **Q2: Rigorously Excluding Textual Knowledge Confounding.**
    The paper asserts that visual priors emerge from text-only training, yet fails to **strictly exclude** that this is simply the encoding of **structured, descriptive visual language** found in the massive text corpus. Have the authors attempted to train a small LLM on a corpus **devoid of all relational and spatial language** to see if the reasoning prior still emerges? If this **linguistic confounding** is not ruled out, the causal foundation of the central thesis—that "LLMs learn to see before seeing"—is a fatal weakness.

3.  **Q3: Transferability Challenge of the P-SET Proxy Tasks.**
    The P-SET tasks are synthetic. While the paper shows a **correlation** between the prior and downstream MLLM performance, how **robust** is this correlation? Have the authors performed an ablation showing that an LLM with a higher P-SET score *consistently* yields superior downstream performance even when subjected to *different* minimal fine-tuning techniques (e.g., LoRA vs. full tuning)? If a high P-SET score does not reliably predict downstream gains, its value as an **efficient MLLM pre-training metric** is questionable.

4.  **Q4: Influence of Specific LLM Family on Scaling Laws.**
    The scaling laws are predominantly based on the Llama family or similar Transformer architectures. Have the authors attempted to replicate the scaling trends of these priors on **non-Transformer architectures** (e.g., Mamba or RetNet)? If these laws are **architecture-dependent** rather than a **universal property of language models**, doesn't the demystification of visual priors only serve as a description of a **specific model family** instead of a **general law of learning**?

---

> ### Author Response · Authors · 2025-11-19
> **Thank you & reply**
>
> We sincerely thank you for your thorough and insightful review! Your questions cut to the core of our work's mechanistic and causal foundations, and we appreciate the opportunity to clarify our results and strengthen our claims. Below, we address each of your concerns and questions in detail.
>
> **W1 and Q1: Mechanistic Evidence for the Separation of Priors.**
>
> We mainly provide strong statistical and behavioral evidence supporting the functional separability of these priors, here are some summaries:
>
> **Statistical independence:** Figure  5 (Finding 5) shows a near-zero correlation between "Perception-heavy" (General/OCR) and "Reasoning-heavy" (Knowledge/Vision-Centric) tasks, indicating they form distinct skill clusters.
>
> **Distinct Origins:** Figure 2 (Finding 3) and  Finding 3 (Figure 4) demonstrate that reasoning priors emerge strictly from reasoning-centric data (Code/Math), whereas perception priors are more diffuse and benefit from diverse web data.
>
> **Universal vs. Dependent:** Finding 6 (Figure 6) shows that the reasoning prior is "universal" (scales with reasoning data regardless of the vision encoder),  while the perception prior is highly sensitive to the choice of vision encoder. Hypothesis 2 (Figure 9, 10, 13, 14) further suggests that language reasoning skills can be directly transferred to solve visual problem
>
> These distinct scaling laws and dependencies provide strong evidence that these priors function as separable components. We argue that the consistent functional dissociation observed across multiple independent axes already provides compelling evidence for their distinct nature, sufficient to support our findings. However, we fully agree that internal representation analysis is a valuable direction, and we leave this exploration to future work on explainability.
>
> **W2 and Q2:  Fatal Causal Confounding of Textual Knowledge and Visual Priors. Have the authors attempted to train a small LLM on a corpus devoid of all relational and spatial language to see if the reasoning prior still emerges?**
>
> As shown in Appendix Table 4 (page 25), reasoning-centric corpora like code and math have a very low density of "Visual Concepts" (3.3% and 7.8%) compared to others like literature.  To further rigorously rule out confounding, we conducted a new experiment training an LLM solely on code data from which all visual terms and concepts were filtered out. Despite zero exposure to explicit visual descriptions, this model still exhibited strong visual reasoning capabilities in the downstream MLLM, achieving 31.8 avg VQA performance and 42.7 avg in vision-centric tasks. Compared to other results shown in Figure 2,  this code data without any visual content still ranks top 3 in VQA performance (out of 17 in total now). This confirms that the visual reasoning prior stems from learning abstract structural and logical patterns.
>
> We would also like to refer to our answers to **Reviewer 1T9q** on question W3 on a related question: **visual priors versus general language capacity.** Where we discussed the relationship between visual prior and language capacity, and we showed that stronger general language capacity does not necessarily lead to better visual performance in controlled settings, effectively isolating the two abilities.
>
> **W3: Lack of Prescriptive Guidance (text pre-training data optimization strategy) for MLLM Design.**
>
> Providing such guidance was a central goal of our work! Our paper explicitly covers quantifiable text pre-training data optimization strategy for MLLM performance, mainly in Finding 4 (Section 3.4) and Section 4 (Scaling up experiments). In Section 3.4, we derive a specific, optimized data mixture (Table 2, mix6) that balances language and vision capabilities. In Section 4, we apply this recipe to train a 7B model on 1T tokens. Table 3 demonstrates that this "Balanced" recipe outperforms the "Language-Favorable" baseline on overall VQA tasks while maintaining strong language performance.

---

> ### Author Response · Authors · 2025-11-19
> **Reply 2**
>
> **Q3: Transferability Challenge of the P-SET Synthetic Proxy Tasks.**
>
> To clarify, we believe our evaluation suite (referred to as "P-SET" in the review) is not synthetic. It consists of 16 standard, comprehensive public benchmarks (e.g., MMBench, ScienceQA, TextVQA, MMMU, BLINK...) widely used to assess real-world MLLM performance (see Section B.2 MLLM evaluation on page 22 for details).
>
> We agree it would be beneficial to test different fine-tuning techniques for visual adaptation to see if they yield similar visual performance for different LLMs.To this end, we compared LoRA (applied during visual SFT to LLM backbone) with full fine-tuning on models pre-trained with five distinct data sources: code, food, web-crawl, literature, and biology (5 models from Finding 2, Figure 2). Previously, these pre-training sources yielded performance in a decreasing order. Our experiment aimed to determine if this trend persists regardless of the fine-tuning method. The results showing avg VQA performance are presented below, in which consistent trends are observed.
>
> | Method | code | food | web-crawl | literature | biology |
> | :--- | :---: | :---: | :---: | :---: | :---: |
> | LoRA | 31.7 | 31.3 | 31.1 | 30.1 | 29.6 |
> | Full | 32.1 | 31.5 | 31.1 | 30.5 | 29.8 |
>
> **Q4: Influence of Specific LLM Family on Scaling Laws.**
>
>
> We agree it would be beneficial to test whether our findings apply to non-Transformer architectures! We are running experiments to this end and will update the results once they are available.

---

> > ### Author Response · Authors · 2025-11-24
> > **Reply 2 (Continue)**
> >
> > **Q4: Influence of Specific LLM Family on Scaling Laws.**
> >
> > We conducted additional experiments using Mamba-2, a representative SSM architecture.
> >
> > Experimental Setup:
> > We replaced the Transformer backbone with Mamba-2 while maintaining the exact same training data and visual adaptation strategy (MLP projector) used in our main experiments. We trained models on the five pre-training domains: code, food, web-crawl, literature, and biology.
> >
> >
> > The average performance on VQA is presented below:
> >
> > | architecture | code | food |web-crawl | literature | biology |
> > | :--- | :---: | :---: | :---: | :---: | :---: |
> > | Llama-3 | 32.1 | 31.5 | 31.1 | 30.5 | 29.8 |
> > | Mamba-2 | 30.5 | 29.6 | 30.0 | 29.1 | 28.7 |
> >
> >
> >
> >
> > The results indicate that our findings regarding data quality transfer effectively to non-Transformer architectures. Similar to the Transformer results, models pre-trained on code achieved the highest performance (30.5), while biology remained the lowest (28.7). While we observe a minor permutation where web-crawl (30.0) slightly outperforms food (29.6), the overall decreasing trajectory is preserved.
> >
> > This suggests that visual prior is general laws in pre-training, rather than only being applied to specific architecture.

---

### Official Review · Reviewer_Zsbw · 2025-11-01

**Soundness:** 4
**Presentation:** 4
**Contribution:** 4
**Rating:** 6
**Confidence:** 4

**Summary:**

This paper investigates the emergence of visual priors in Large Language Models (LLMs) that are trained solely on text. these text-only models encode surprisingly rich visual understanding, which can be unlocked through lightweight multimodal adaptation. The authors conduct over controlled experiments that systematically vary model scale, pre-training data composition, and adaptation strategies to dissect the structure and origin of such priors. Their key findings include:

1. Visual priors decompose into perception and reasoning components
2. Reasoning priors are predominantly acquired from text sources involving code, math, and academic language
3. Perception priors emerge diffusely from broad web-crawl text and depend heavily on visual instruction tuning

Lastly, the paper introduces MLE-Bench, a benchmark focused on perception-specific evaluation.

**Strengths:**

1. The decomposition of visual priors into perception and reasoning is novel and theoretically impactful; it challenges assumptions of monolithic visual semantics in LLMs.
2. Extensive and well-structured experimental setup across five model sizes, varying data domains, and different visual encoders (MetaCLIP, DINOv2, MAE).
3. Offers practical insights for data-centric pre-training of future MLLMs—highly relevant to both industry and academic development.
4. The paper conducts over 100 controlled pretraining runs, systematically varying data mixtures, encoder choice, and instruction-tuning strategies.
5. The paper is well-written and structured logically.

**Weaknesses:**

1. The paper’s split of pretraining text into “reasoning” vs. “visual” buckets hinges on an automated pipeline: a 32B dense LLM is prompted to multi-label 1024-token segments into categories such as visual concept/attribute/relationship and code/math/science reasoning; those labels then drive mixture sweeps and downstream claims about where “priors” come from. That setup introduces several risks.
First, it is model-dependent: is there consistency across other models? i.e. other than Qwen
Second, the error characteristics are unknown: there was no report of inter-annotator (human) checks are reported for the classifier, so mislabeling rates—especially on ambiguous text that mixes descriptive visual language with abstract reasoning—are unconstrained
My main concern here is whether the LLM-based labeling truly distinguishes “visual” from “reasoning” content, or if the observed gains simply reflect differences in data quality—i.e., one category inadvertently contains cleaner or more informative text, which could artificially boost general VQA performance.

2. The paper’s validation at scale is constrained and, in some cases, internally inconsistent, which makes it difficult to confidently generalize the findings beyond the mid-scale (30–50B token) regime. While the main conclusions are derived from extensive 30–50B-token experiments, the authors extend only two mixtures to 1T tokens, both using a 7B LLM: one “language-favorable” (mix 0) and one “balanced” (mix 6). Here, the results in Table 2 (mid-scale) show that mix 0 outperforms mix 6 on pure language benchmarks, as expected from its language-heavy composition. However, in Table 3 (1T-token scale), the trend reverses on several language tasks—mix 6 overtakes mix 0 despite having proportionally less “language” text. This reversal is puzzling and suggests a possible interaction between model scale, token budget, and mixture composition that is not fully explored. Could the authors elaborate more on this?

Compounding this, the 1T-token budget far exceeds the Chinchilla optimal region for a 7B model (≈150–200B tokens), placing both scaled-up models deep into the over-training regime where diminishing return are likely. That makes it difficult to attribute performance shifts solely to mixture content, rather than to scaling artifacts or excessive token exposure.

**Questions:**

1. The perception vs reasoning tasks showed weak or negative correlation (Figure 4). Can the authors clarify why such inverse relationships might occur?
2. since hallucination "stronger models do not necessarily guarantee fewer hallucinations in this blind VQA", did the authors measure hallucination rates systematically?

---

> ### Author Response · Authors · 2025-11-19
> **Thank you & reply**
>
> We are very encouraged to have received three excellent (4) ratings on Soundness, Presentation, and Contribution from you! We sincerely thank you for the constructive assessment and for raising several insightful and meaningful questions.
>
> **W1: Consistency across other models? Whether the LLM-based labeling truly distinguishes “visual” from “reasoning” content? Whether the observed gains simply reflect differences in data quality?**
>
> **Consistency:** To investigate this, we performed a new experiment. We used a Llama-3.3 70B Instruct model to re-classify a roughly 50B-token subset of our web-crawl data using the same prompt and methodology. We found a high degree of consistency with our original labels. The label agreement for the reasoning combination category was ~89%, and for the visual combination category, it was ~91%, they both show high consistency, probably because the prompt template is clear and the task is not hard.
>
> **Distinguishes “visual” from “reasoning”:** Our pipeline was designed to handle ambiguity by performing multi-label classification. A single text segment can be assigned both reasoning and visual labels if it contains elements of both. This mitigates the risk of forcing a difficult choice on ambiguous text. As shown in Table 4 (page 25), the data sources that are strong drivers of reasoning priors have very little overlap with visual content, and vice-versa. For instance, the code corpus is classified as 96.7% code reasoning but only 3.7% visual combination. This indicates that the amount of highly ambiguous text that the classifier must adjudicate is relatively small, as the source domains are themselves quite distinct, thus further reducing the potential for such mislabeling.
>
> **Data quality:** We can check this by experiments that do not rely on the LLM classifier. In our single-source pre-training (Finding 2, Figure 3), models trained on entire, unfiltered sources like code and math demonstrated superior reasoning-centric VQA performance. This shows that the inherent nature of a data source cultivates specific visual priors. Furthermore, our mixture experiments (Finding 4, Table 2) blended these complete sources, not based on LLM-filtered segments. The consistent VQA improvements from adding more reasoning-heavy sources confirm that the observed effects stem from the type of data.
>
> **W2:  There is a possible interaction between model scale, token budget, and mixture composition that is not fully explored.**
>
> We posit that this reversal is attributable to an interaction between data composition and training scale. Our hypothesis is that for the benefits of reasoning-heavy pre-training to fully manifest in core language abilities, the model must first be grounded in a sufficiently large volume of foundational text. In our 1T-token experiments, we observed that this reversal point occurred after approximately 600B tokens of training. Before this threshold, mix0 consistently led mix6 in language proficiency. This suggests that with a sufficiently large training budget, the structured knowledge from reasoning-centric data, once properly grounded, can contribute more significantly to overall language competence.
>
>
> **W3: 1T-token budget far exceeds the Chinchilla optimal region for a 7B model.**
>
> 1T-token runs for a 7B model far exceed the compute-optimal budget suggested by Chinchilla scaling laws (~150-200B tokens). Our intention was to follow the common practice of frontier models (e.g., Llama-3 with 15T tokens, Qwen-3 with 36T tokens), which often prioritize final model performance over training efficiency.
>
> We agree that validation within the optimal region is valuable. We have run an additional experiment training our 7B models on mix0 and mix6 for 200B tokens. The results are presented below:
>
> | Model | ppl | Language avg acc | General | Knowledge | OCR & Chart | Vision-Centric | VQA avg |
> | :--- | :---: | :---: | :---: | :---: | :---: | :---: | :---: |
> | Language-Favorable | 10.41 | 0.622 | 44.98 | 27.50 | 20.87 | 45.11 | 34.62 |
> | Balanced | 10.88 | 0.614 | 46.41 | 27.92 | 22.64 | 45.45 | 35.61 |
>
> Our central conclusion with the visual priors remains robust, where the Balanced model continues to yield higher overall VQA performance, outperforming Language-Favorable across all categories.

---

> > ### Author Response · Authors · 2025-11-19
> > **Reply 2**
> >
> > **Q1: The perception vs reasoning tasks showed weak or negative correlation.**
> >
> > Our correlation analysis is based on 105 results from 3B models. The majority of these models (81 out of 105) originate from the controlled mixing experiments in Sections 3.3 and 3.4. In these experiments, the pre-training data mixtures were often intentionally and heavily skewed towards reasoning-centric content.
> >
> > Pre-training a model on a data mixture predominantly composed of reasoning sources like code and math within a fixed token budget (30B tokens) naturally comes at the expense of exposure to broad, descriptive corpora. Consequently, while these models develop a reasoning prior, they may lack the diverse descriptive vocabulary essential for fine-grained perceptual tasks. This specialization, driven by our experimental design to probe the origins of priors, is the primary driver of the weak inverse relationship shown in the correlation matrix.
> >
> >
> > **Q2: Measure hallucination rates systematically.**
> >
> > Thank you for carefully reading our manuscript, including this part in the Appendix that was not mentioned in the main paper, we truly appreciate your time and helpful suggestions.
> >
> > We did not perform a systematic, quantitative measurement of hallucination rates across all models. A systematic study of this phenomenon would require evaluating it across a large, controlled set of prompts for many models. While this is a critical and valuable direction for future research, it was beyond the primary scope of our paper.
> >
> > Here, we present a preliminary attempt to study this. We tested Gemini-2.5-Pro on 400 different prompts asking for question answers, consisting of 200 multiple-choice questions and 200 open-world VQA questions. No images were provided to the model. We used a temperature of 1.0 to encourage diverse outputs. For the multiple-choice questions, the hallucination rate was 74.5%. In contrast, it was only 8% for the open-world VQA questions.
> >
> > We hypothesize this discrepancy is due to the nature of the model's RL post-training. Models that undergo extensive RL post-training are often rewarded for selecting an answer from a provided list. When faced with a multiple-choice format but no image, the model's strong prior is to 'play the game' and choose an option, even without visual evidence, leading to a high hallucination rate. For open-world questions, however, the model is not forced into a specific response format. This allows it to more frequently default to a safer, more truthful response, such as stating that it cannot see an image.

---

> > > ### Comment · Reviewer_Zsbw · 2025-11-20
> > >
> > > Thank you for the detailed responses. My concerns have been largely resolved, I've updated my score!

---

> > > > ### Author Response · Authors · 2025-11-24
> > > >
> > > > We are thrilled to hear that your concerns were addressed and thank you again for your insightful comments!!

---

### Author Response · Authors · 2025-11-19
**General response**

We are deeply grateful to all four reviewers for their thoughtful and constructive feedback! We are delighted that all reviewers recognized the importance and pioneering nature of our work (**All**), its novelty (**R-Zsbw, R-fgou**), its extensive and rigorous experiments (**R-Zsbw, R-1T9p**), its practical value for the MLLM community (**R-Zsbw, R-JSpZ**), and the clarity of the presentation (**R-Zsbw, R-JSpZ**).

We have uploaded our revised manuscript and addressed individual questions in separate threads. Thank you again for your time, and we look forward to further discussions!

---

### Meta-Review · Area_Chair_YdvN · 2026-01-06

**Summary:**

This paper investigates how text-only LLMs develop visual priors without image exposure, distinguishing between perception and reasoning priors through extensive experiments (100+ experiments, 500K GPU hours). Reviewers acknowledged the novelty and thoroughness of the work. The rebuttal comprehensively addressed all major concerns with substantial new experiments, leading to a strong consensus (final scores: 6->8, 6, 8, 8). Accept.

**Reviewer Concerns:**

Addressed by rebuttal:

- Data labeling validity: Re-verified with Llama-3.3 70B, confirming pipeline reliability.

- Architectural generalization: Validated on Mamba-2 (SSM), showing findings hold beyond Transformers.

- Textual confounding: Code-trained model with filtered visual terms still developed visual reasoning, ruling out mere linguistic correlation.

- Fine-tuning generalization: LoRA vs. full fine-tuning comparison confirmed robustness.

Outstanding concerns:

None significant. All major concerns were substantively addressed with new experiments.

**Reviewer Scores:**

The initial ratings were 6, 6, 8, 8, and were subsequently updated to 6, 8, 8, 8.

Zsbw 6->8 Explicitly raised score after concerns resolved

JSpZ 8->8 Confirmed questions addressed

fgou 6-> 6 No response

1T9p  8->8 No response

---

### Decision · Program_Chairs · 2026-01-26

Accept (Oral)